# Clinical Interest of Serum Alpha-2 Macroglobulin, Apolipoprotein A1, and Haptoglobin in Patients with Non-Alcoholic Fatty Liver Disease, with and without Type 2 Diabetes, before or during COVID-19

**DOI:** 10.3390/biomedicines10030699

**Published:** 2022-03-17

**Authors:** Olivier Deckmyn, Thierry Poynard, Pierre Bedossa, Valérie Paradis, Valentina Peta, Raluca Pais, Vlad Ratziu, Dominique Thabut, Angelique Brzustowski, Jean-François Gautier, Patrice Cacoub, Dominique Valla

**Affiliations:** 1BioPredictive, 75007 Paris, France; olivier.deckmyn@biopredictive.com (O.D.); valentina.peta@biopredictive.com (V.P.); 2Assistance Publique-Hôpitaux de Paris (AP-HP), Department of Hepato-Gastroenterology, Pitié-Salpêtrière Hospital, 75013 Paris, France; raluca.pais@aphp.fr (R.P.); vlad.ratziu@inserm.fr (V.R.); dominique.thabut@aphp.fr (D.T.); 3Sorbonne Université, INSERM, Centre de Recherche Saint-Antoine (CRSA), Institute of Cardiometabolism and Nutrition (ICAN), 75013 Paris, France; 4Department of Pathology, Physiology and Imaging, Beaujon Hospital APHP Diderot University, 75006 Paris, France; pierre.bedossa@inserm.fr (P.B.); valerie.paradis@aphp.fr (V.P.); 5Assistance Publique-Hôpitaux de Paris (AP-HP), Department of Hepatology, Beaujon Hospital, 92110 Clichy, France; angelique.brzustowski@inserm.fr (A.B.); dominique.valla@aphp.fr (D.V.); 6Inserm U1149, Centre de Recherche sur l’ Inflammation CRI, 92110 Clichy, France; jean-francois.gautier@aphp.fr; 7Department of Diabetes and Endocrinology, Lariboisière Hospital APHP, Université de Paris, 75007 Paris, France; 8Assistance Publique-Hôpitaux de Paris (AP-HP), Department of Internal Medicine and Clinical Immunology, Pitié-Salpêtrière Hospital, 75013 Paris, France; patrice.cacoub@aphp.fr; 9Département Hospitalo-Universitaire I2B, Sorbonne Université, 75006 Paris, France; 10UMR 7211 (UPMC/CNRS), UMR S-959 (INSERM), 75006 Paris, France

**Keywords:** alpha-2 macroglobulin, apolipoprotein A1, haptoglobin, type 2 diabetes, non-alcoholic fatty liver disease NAFLD, non-alcoholic steatohepatitis NASH, SAF-scoring system, liver fibrosis, steatosis, COVID-19

## Abstract

In patients with non-alcoholic fatty liver disease (NAFLD) with or without type 2 diabetes mellitus (T2DM), alpha-2 macroglobulin (A2M), apolipoprotein A1 (ApoA1), and haptoglobin are associated with the risk of liver fibrosis, inflammation (NASH), and COVID-19. We assessed if these associations were worsened by T2DM after adjustment by age, sex, obesity, and COVID-19. Three datasets were used: the “Control Population”, which enabled standardization of protein serum levels according to age and sex (N = 27,382); the “NAFLD-Biopsy” cohort for associations with liver features (N = 926); and the USA “NAFLD-Serum” cohort for protein kinetics before and during COVID-19 (N = 421,021). The impact of T2DM was assessed by comparing regression curves adjusted by age, sex, and obesity for the liver features in “NAFLD-Biopsy”, and before and during COVID-19 pandemic peaks in “NAFLD-Serum”. Patients with NAFLD without T2DM, compared with the values of controls, had increased A2M, decreased ApoA1, and increased haptoglobin serum levels. In patients with both NAFLD and T2DM, these significant mean differences were magnified, and even more during the COVID-19 pandemic in comparison with the year 2019 (all *p* < 0.001), with a maximum ApoA1 decrease of 0.21 g/L in women, and a maximum haptoglobin increase of 0.17 g/L in men. In conclusion, T2DM is associated with abnormal levels of A2M, ApoA1, and haptoglobin independently of NAFLD, age, sex, obesity, and COVID-19.

## 1. Introduction

For more than a century, diabetes has been associated with liver disease and severe pneumonia [1]. Recently, the COVID-19 pandemic revealed that patients with type 2 diabetes (T2DM), non-alcoholic fatty liver disease (NAFLD), or metabolic liver disease were at higher risk of being infected with and hospitalized for moderate-to-severe COVID-19 complications than control patients without T2DM or NAFLD [2,3,4,5,6]. However, the causal relationship between COVID-19 susceptibility/severity and NAFLD remains unclear because of confounders such as age, sex, obesity, T2DM, stage of fibrosis, and grade of inflammation, also called non-alcoholic steatohepatitis (NASH) [6,7,8].

Three ubiquitous serum proteins that are involved in cell repair and immunity, alpha-2 macroglobulin (A2M) [9,10,11,12], apolipoprotein A1 (ApoA1) [9,10,13,14], and haptoglobin [9,10,15,16,17,18], which are mainly synthetized by the liver, are associated with the risk of being infected with COVID-19 and are associated with the severity of its complications compared with controls without diabetes nor NAFLD [9,10,11,12,13,14,15,16,17,18]. These proteins are also associated with liver fibrosis and inflammatory activity in the most frequent liver diseases, including NAFLD [19,20,21,22,23,24,25,26,27,28,29,30,31,32,33,34,35,36,37,38,39,40,41,42,43,44,45,46,47,48,49,50,51,52,53,54,55,56,57,58,59,60,61].

These serum proteins are easy to assess and could be used as biomarkers of the risk of SARS-CoV-2 infection, particularly in patients with T2DM and NAFLD, a large part of the global population that has a higher risk of SARS-CoV-2 infection [9,10].

The epidemiological purpose of this work was to assess the impact of T2DM on the concentrations of these proteins in the blood according to the main confounders of age, sex, and obesity, and the severity of three liver features, fibrosis, NASH inflammatory activity, and steatosis.

The clinical purpose of this work was to prevent misinterpretation and false positives and negatives of the serum levels of such major proteins observed in patients with metabolic liver disease who express T2DM with or without SARS-CoV-2 infection. Increasing numbers of new or known proteins were included in multivariate diagnostic/prognostic tests and analyzed. In simple terms, a serum level of 3.8 g/L A2M can be observed in an asymptomatic 18-year-old girl or in an obese 60-year-old man with cirrhosis, but should not be interpreted similarly.

Because of their highly conserved evolutionary ubiquitous properties, it is logical that these three proteins are associated with three global human diseases, T2DM, COVID-19, and liver fibrosis [20,21,22,23].

A2M, a glycoprotein of 720 kDa mass, is a major component in the circulation of vertebrates. It belongs to a family of extracellular matrix regulators. As well as the rapid inhibition of proteinases released during inflammation, A2M has many functions [19,23]. As stated by Rehman et al.: “This multipurpose antiproteinase is not ‘‘fail safe’’ and could be damaged by reactive species generated endogenously or exogenously, leading to various pathophysiological conditions” [19]. Indeed, in children and adults younger than 25 years old, A2M expression is significantly higher than in older healthy adults [24] and is associated with a lower risk of several severe diseases such as severe COVID-19, *Trypanosoma cruzi* infection (Chagas disease) [25], and intervertebral disc degeneration [26].

In COVID-19, a possible protective role of elevated native A2M in children was recently identified and deserves more in-depth scientific exploration [12]. 

In T2DM, increased A2M was first described in 1967 [27], both in adults and children with very early T2DM [27,28], suggesting very early damage by the protein [19,27]. 

In chronic liver disease, increased A2M has been described since 1963 and is associated with liver fibrosis, [29,30]. A2M has been successfully included in multivariate analyses of fibrosis biomarkers, mostly combined with ApoA1 and haptoglobin, for the surveillance of chronic viral hepatitis C and B, alcoholic liver disease, and various metabolic liver diseases including dyslipidemia, NAFLD with and without T2DM [31,32], sole T2DM [27,28,32,33,34,35,36,37], steatosis [33], and non-alcoholic steatohepatitis (NASH) [34,35,36,37], and has been associated with severe obesity [39], patients with bariatric surgery [39], obstructive sleep apnea [40], and the general population [31]. 

ApoA1, a 45.4 kDa mass protein, is mostly found in association with the high-density cholesterol (HDL) moiety. In addition to its role in regulating cholesterol and protecting against cardiovascular disease, ApoA1 has many functions in inflammatory and immune responses. ApoA1 inhibits apoptosis and pro-oxidative and proinflammatory processes in endothelial cells, induces vasodilation, inhibits the activation of platelets, and contributes to innate immunity [41].

In severe pneumonia, the association between low HDL and severe pneumonia has been well established for more than a century [9], and the lowest levels have been observed in patients with moderate and severe COVID-19 with significant prognostic value [9,10,42,43,44,45,46,47,48,49]. There is a causal association for subjects with low ApoA1 and, 10 years later, low levels were correlated with the risk of COVID-19 [13]. 

In T2DM, decreased ApoA1 was first described in 1982 [46]. Downregulated ApoA1 in T2DM could be related to damage of the protein and loss of the protective functions of the native protein including three post translational modifications: oxidation, carbamylation, and glycation [46,47,48,49].

In chronic liver disease, the decrease in ApoA1 according to the progression of liver fibrosis was first described in 1986 [50,51], and has been successively included in multivariate biomarkers, mostly combined with A2M and haptoglobin, for the surveillance of chronic liver diseases including NAFLD and NASH with or without T2DM [32,33,34,35,36,37,38,52]. In a large cohort of patients at risk of NAFLD in the USA, the “NAFLD-Serum” cohort, there was a significant decrease in ApoA1 during the first wave of the 2020 pandemic compared with the respective months in 2019 [9]. ApoA1 is mainly synthetized in the liver and intestine. In liver fibrosis, decreased serum ApoA1 is observed without hepatic insufficiency, the ApoA1 being trapped by the collagenization of the endothelial cells [51] before advanced fibrosis and before the hepatic insufficiency. In patients with cirrhosis, ApoA1 and HDL3 levels were significantly lower in patients who developed severe infection [53]. This sensitivity of ApoA1 is an advantage for its inclusion in multivariate biomarkers because of its prognostic value.

Another possible source of major ApoA1 variability is that enterically derived HDL restrains liver injury through the portal vein, with ApoA1 inhibiting the bacterial lipopolysaccharide (LPS) source of inflammation [53]. The biogenesis of HDL requires ApoA1 and the cholesterol transporter ABCA1. Although the liver generates most of the HDL in the blood, HDL synthesis also occurs in the small intestine. The intestine produces the small form of HDL called HDL3 that it is enriched in lipopolysaccharide (LPS)-binding protein (LBP). In complexes with LBP, HDL3 prevented LPS-binding and the inflammatory activation of liver macrophages [53]. Indeed, ApoA1 and HDL3 have strong translational potential in the understanding of T2DM, NASH, and severe infection. 

Haptoglobin is synthesized predominantly in hepatocytes as a single 45 kDa polypeptide that rapidly forms dimers through disulfide bond formation [54]. The α-chain of haptoglobin is then proteolytically cleaved, resulting in the final tetrameric form, Hp 1-1. One of the most striking effects of the haptoglobin polymorphism resides in its structural heterogeneity, with the existence of haptoglobin in different oligomeric states depending on its genotype: Hp1-1, Hp2-1, or Hp2-2. Haptoglobin has a significant role in clearing toxic hemoglobin (Hb) through high-affinity binding to the macrophage scavenger receptor CD163. In addition to this antioxidant function, haptoglobin has a role in the immune response and during the acute phase response. In healthy subjects, haptoglobin concentrations in the circulation are very low. Haptoglobin reduces the loss of free Hb through glomerular filtration, controlling heme detoxification and promoting iron recycling. Haptoglobin can act either as an anti-inflammatory modulator or as a pro-inflammatory activator suppressing T cell proliferation and regulating the balance between T helper cells Th1 and Th2. Hp1-1:Hb complexes induce Th2 cell-dependent pathways that allow healing and repair. Conversely, macrophages activated by Hp2-2:Hb stimulate the Th1 response that promotes pro-inflammatory cytokines. Haptoglobin is a marker of inflammation, its level increasing during infections, injuries, and malignancies. The functional properties of haptoglobin reflect the different phenotypes Hp1-1, Hp2-1, and Hp2-2. Hp2-2 is the less active form of haptoglobin, and patients with this phenotype show a significantly higher risk of cardiovascular, neurological, and infectious complications compared with Hp1-1 and Hp2-1 individuals [15,55].

In patients with COVID-19, haptoglobin phenotypes (Hp1 and Hp2 alleles) were not associated with mortality [56]. Regardless of the phenotype, serum haptoglobin in humans and non-human primates is elevated very early, at 2 days after SARS-CoV-2 infection, and for a longer time than C-reactive protein (CRP) [9,10]. 

Patients with T2DM and Hp2-2 phenotypes are at a significantly higher risk of microvascular and macrovascular complications via obesity effects in the Mexican population [56,57]. To date, a direct causal association between haptoglobin phenotypes and the occurrence of T2DM has not been established [56,57,58]. In the progression of asymptomatic obesity to a T2DM status, the increased influx of immune cells, especially macrophages, into visceral adipose tissue is mediated by adipocyte-derived chemokines, and this influx is accompanied by inflammatory cytokines such as tumor necrosis factor α (TNFα) and interleukin 6 (IL6) [59,60]. Haptoglobin related protein (HPR) mRNA expression is significantly increased when comparing healthy obese individuals with impaired glucose fasting obese patients and obese patients with T2DM [15]. 

In patients at risk of NAFLD, haptoglobin is increased in obese patients with T2DM but decreased in patients who progress to advanced fibrosis [61]. No study thus far has assessed the respective impacts of elementary histological features (steatosis, NASH inflammatory activity, and fibrosis) adjusted according to T2DM and obesity on serum haptoglobin.

Here, the aim was first to standardize the three protein values according to age and sex, two major confounding factors, using published normal values from a healthy general population in the USA, called here the “Control Population” cohort. 

Second, we assessed the impact of T2DM on these proteins according to the main metabolic liver features of fibrosis and inflammation (NASH) and steatosis without inflammation, stratified by obesity. We used NAFLD patients with liver biopsies who were centralized and analyzed using the validated scoring systems (SAF), called the “NAFLD-Biopsy” cohort. 

Third, we assessed the impact of T2DM on these proteins in “NAFLD-Serum” patients followed before and during the COVID-19 pandemic, adjusted according to the confounders of age, sex, and obesity.

We found that in patients at risk of NAFLD, the impact of T2DM on these three proteins should be studied not only after standardization according to age and sex, but also after stratification by obesity. In the two cohorts, the three proteins levels were significantly different than the normal values. In patients at risk of NAFLD without T2DM, A2M was increased, ApoA1 was decreased, and haptoglobin was increased. In patients with both NAFLD and T2DM, these significant differences were magnified. During SARS-CoV-2 infection, this population acquire a third factor of decreasedApoA1 and increased haptoglobin. These results were in line with the independent diagnostic and prognostic values of ApoA1 in COVID-19. 

## 2. Materials and Methods

### 2.1. Study Participants and Design

This retrospective non-interventional epidemiological study had three co-primary aims (Figure 1). 

### 2.2. Standardization of Protein Values

To standardize the three protein values, we used the reference values of studies based on the “Control Population” cohort of 40,420 Caucasian individuals from northern New England. Sera were assessed between 1994 and 2000. Measurements were standardized against Certified Reference Material for Proteins in Human Serum (RPPHS), and the results were analyzed using a previously described statistical approach. Individuals with unequivocal laboratory evidence of inflammation, CRP > 10 mg/L defining significant acute phase reactant (APR), were excluded in one leg of the study and included in the other, confirming that A2M does not respond to acute phase drive in humans [24]. Most samples were sent for study by physicians because of a suspected diagnosis or symptom. Computer processing of over 28,000 unique diagnostic strings required that they be classified into 93 categories representing related conditions. Diagnostic codes containing individuals with conditions expected to have a direct effect on serum protein levels, for example multiple myeloma, cirrhosis, hepatitis, infection, lung disease, leukemia, renal failure, rheumatic disease, and immunodeficiency, were excluded. Outliers were identified in this way among the diagnostic group parameters associated with codes not expected to alter serum protein values. A logarithmic transformation of the variance was corrected for skewness. The resulting trimmed mean value, ±1.96 standard deviation for both the multiples of the median (MoM) and the log variance, defined the limits of acceptability. Measurements from any diagnostic group falling within the limits were considered reference values, and those falling outside were not.

#### Statistical Analysis

Here, we applied the methodology previously published by Ritchie et al. [24,62,63,64]. For studying proteins, 28,239, 28,919, and 27,382 cases were available without significant APR for A2M [24], ApoA1 [63], and haptoglobin [64], respectively. In this “Control Population”, there were 225 cases with a diagnosis of T2DM and no APR, and 95 with APR, and both were outside the reference range for log MoM and log variance. The proposed method of converting laboratory results to multiples of the age- and sex-specific MoMs has several advantages. Conversion permits each analyte to fit a logarithmic Gaussian distribution reasonably well, allowing each MoM level to be assigned a centile. Thus, a laboratory measurement can not only be reported in mass units but also through conversion to MoM, the associated centile based on that individual’s age and sex. The regression models and coefficients for median proteins measurements by age and sex have been described previously [24,63,64].

### 2.3. Impact of T2DM on Proteins According to Histological Metabolic Liver Features

#### 2.3.1. Patients

Two previously published datasets that validated the diagnostic performances of blood biomarkers were retrospectively integrated in the present “NAFLD-Biopsy” cohort [37,52] (Figure 1). One was the construction and internal validation performed in the fatty liver inhibition of progression (FLIP) and FibroFrance cohorts [52], and the other was the external validation in the QUIDNASH prospective cohort [37]. Details were provided in each publication and summarized in Appendix A with the list of participants and the projects’ summaries. All these clinical non-interventional studies were approved by the ethics committee at each participating institution and were performed according to good clinical practice and the Declaration of Helsinki, and all patients provided written informed consent.

The FLIP project is supported by the European Community’s Seventh Framework Program (FP7/2007-2013) under grant agreement number HEALTH-F2-2009-241762. FibroFrance is supported by the National Clinical Research (CPP-IDF-VI, 10-1996-DR-964, DR-2012-222) and declared in the Clinical Registry (number: NCT01927133).

The QUIDNASH study, NCT03634098, was approved by the Research Ethics Committee (#18.021-2018-A00311-54). In patients with T2DM, the diagnostic accuracy of FibroTest, NashTest-2, and SteatoTest-2 was assessed using liver histology as the reference to evaluate liver fibrosis, NASH, and steatosis, and is detailed elsewhere [39]. Briefly, NAFLD was suspected on the basis of the presence of abnormal liver enzymes as well as an ultrasound scan showing a bright liver echo pattern in patients with T2DM diagnosed at a diabetology outpatient clinic. Consecutive patients were prospectively recruited between October 2018 and 2020 at four outpatient diabetology clinics in the Assistance-Publique-Hopitaux-de-Paris. All patients gave written informed consent. The study was performed in accordance with the Declaration of Helsinki. All authors had access to the study data and reviewed and approved the final manuscript. The chosen same sample size of n = 300 for the primary aim of the study was the same as that used for the internal validation of SteatoTest-2 and for validation of the original SteatoTest. To increase the power of the present study, we added 57 T2DM cases who shared the same inclusion criteria as the QUIDNASH non-interventional cohort after the end of the validation study to the original 272 cases [39].

#### 2.3.2. Blood Tests

A2M, ApoA1, haptoglobin, and fasting glucose were assessed in fresh samples from FibroFrance and QUIDNASH patients. For FLIP patients, serum stored at −80 °C was sent to the reference center, the Biochemistry Department at Pitié-Salpêtrière Hospital, Paris, France.

#### 2.3.3. Histological Reference

The SAF activity scoring system was considered as the simplified histological reference for NASH without the requirements used for NASH-CRN and the FLIP algorithm definition [65,66,67]. The histological references for significant metabolic liver disease (NAFLD) were those defined by the SAF scoring system, fibrosis stage ≥ 2 and activity grade ≥ 2 [66,67]. The goal of the SAF score was to identify a compromise between the development of a simple, easily applied system to make a firm diagnosis in individual patients, even when applied by non-specialists, and of a more reliable and discriminating system for therapeutic trials or for the assessment of biomarker diagnostic performance. A FLIP Histopathology Consortium of eight members developed the FLIP algorithm, a diagnostic tool for the diagnosis and staging of severe forms of NAFLD according to the combination of each semi-quantification of the three elementary features of NAFLD using the SAF score for steatosis SAF-S, inflammatory activity SAF-A, and fibrosis SAF-F. The use of the SAF-A score leads to the selection of patients with more severe disease activity and fibrosis, as observed in a recent trial in NASH [68] in which 76% of patients had significant (F2) or advanced (F3) fibrosis even though no inclusion criterion, with respect to fibrosis stage, was set except for the exclusion of patients with cirrhosis. The use of the SAF-A score enriched the trial, with patients more likely to benefit from pharmacological treatment. As for the other histological end points, the validity of the SAF-A score to define the primary end point as a surrogate for long term outcomes warrants further study [67]. The steatosis score (S) assesses the quantities of large-sized or medium-sized lipid droplets, with the exception of foamy microvesicles, and rates them from 0 to 3 (S0: <5%; S1: 5–33%, mild; S2: 34–66%, moderate; S3: and >66%, marked). Activity (NASH) grade (A, from 0 to 4) is the unweighted addition of hepatocyte ballooning (0–2) and lobular inflammation (0–2). Patients with A0 (A = 0) had no activity; patients with A1 (A = 1) had mild activity; patients with A2 (A = 2) had moderate activity; patients with A3 (A = 3) had severe activity; and patients with A4 (A = 4) had very severe activity. Fibrosis stage (F) was assessed using the following scoring system: stage 0 (F0), none; stage 1 (F1), 1a or 1b perisinusoidal zone 3 or 1c portal fibrosis; stage 2 (F2), perisinusoidal and periportal fibrosis without bridging; stage 3 (F3), bridging fibrosis; and stage 4 (F4), cirrhosis. To reduce interobserver variability and homogenize the reading using the SAF-FLIP histological classification, we used only reports reviewed by members of the FLIP Pathology Consortium (DT and PB for FLIP and FC for the FibroFrance subset), and VP, PB, and BT for the QUIDNASH subset.

#### 2.3.4. Statistical Analysis

We compared the levels of each protein in patients with or without T2DM, defined as fasting glucose ≥ 7 mmol/L [69], versus their expected values in the general population [24,62,63,64] to test the hypothesis that early low levels of A2M and ApoA1 before the onset of T2DM and NAFLD could explain an intrinsic fragility. To account for obesity, these curves were also compared in patients with or without obesity (Figure 1) according to the possible histological confounders, each of them grouped into three classes, fibrosis F2F3F4, NASH A2A3, and steatosis S2S3. For each protein, the curves were assessed by regression according to age separately for women and men, one in patients and one in their expected normal controls. The Loess method was used with 95% confidence intervals and comparisons between curves used the unpaired t-test. 

Univariate correlation matrices were used to assess the correlation between the three proteins and the four confounders in four subsets, with and without obesity, in women and men, called here “sex/obesity” (female/non-obese, male/non-obese, female/obese, and male/obese). A significant association was defined as *p* < 0.05 adjusted by the number of covariables according to the Holm method [70].

Independent associations between the presence of T2DM and the three proteins were assessed by logistic regression analysis in the four subsets, including the three histological features of the SAF scoring system, the five stages of fibrosis (F0 to F4), the four grades of NASH (A0 to A3), and the four grades of steatosis (S0 to S3). This analysis allowed adjustment of the association between T2DM and each protein, independently of the two other proteins of interest after taking into account age and the SAF classes of each feature in each of the four sex/obesity subsets. According to significant correlations between the three features, it seemed fair to perform these regressions separately for each feature. R software was used for the analyses.

### 2.4. Impact of T2DM on the Serum Proteins Levels According to Obesity

#### 2.4.1. Patients

We used both the “NAFLD-Biopsy” cohort and the “NAFLD-Serum” cohort, a large laboratory US cohort from anonymous subjects at risk of liver fibrosis, followed by FibroTest (also known as FibroSure in USA) [9,71]. 

#### 2.4.2. Blood Tests

A2M, ApoA1, haptoglobin, and fasting glucose were assessed in fresh samples, and all laboratories used the methods of the manufacturer of FibroTest [31]. 

#### 2.4.3. Statistical Analysis

The same statistical methods were used as those assessing the impact of T2DM on histological features. Thanks to the sample size, we focused on the impact of T2DM on the three proteins in the four confounder subsets of sex/obesity. 

First, we analyzed the impact of T2DM on proteins stratified by obesity and sex in the “NAFLD-Serum” cohort. Second, we analyzed the univariate correlations between proteins and histological features inside the four sex/obesity subsets. Third, we analyzed the multivariate correlations by logistic regression between proteins and histological features inside the four sex/obesity subsets. Fourth, we analyzed to correlation between the serum level for each protein with BMI, using 5 groups according to WHO definition (18.5, 25, 30, 35 and above 40 kg/m^2^) in all patients and stratified by age in 2 groups (below and equal or above 50 years old).

### 2.5. Impact of T2DM on the Three Proteins According to SARS-CoV-2 Infection

#### 2.5.1. Patients

The “NAFLD-Serum” cohort [9,71] was used.

#### 2.5.2. Blood Tests

A2M, ApoA1, haptoglobin, and fasting glucose were assessed in fresh samples according to the recommendations of the manufacturer of FibroTest [31].

#### 2.5.3. Statistical Analysis

Three methods were used. First, the comparison of proteins levels was performed before and during the COVID-19 pandemic according to T2DM, standardized by age and sex and stratified by obesity. Second, for each patient, the difference between the observed value of the three proteins and the expected normal value adjusted for age and sex were observed. The observed 7-day rolling mean of these differences were compared between 2019 (before the pandemic) and the pandemic period from January 2020 to February 2022.

A figure was built using both the “NAFLD-Serum” dataset and the public data from John Hopkins University (JHU) [72]. The curves of the protein values from January 2019 to 30 January 2022 of the “NAFLD-Serum” dataset, stratified by sex, were graphically compared with those of the JHU dataset regarding mortality and the hospitalization rate, which were only available since March 2020 and August 2020, respectively, and without sex stratification available. 

## 3. Results

### 3.1. Standardization of Proteins Values in the Studied Cohorts Using the “Control Population”

For each age group, the mean difference (%95CI) between the patient protein value and the expected normal value (for the respective age and sex), with its significant *p*-value is displayed at top of each figure. The significance between non-T2DM vs. T2DM patients is displayed in color between the two panels. Characteristics of all “NAFLD-Biopsy” and “NAFLD-Serum” are shown in Table 1, and the subsets according to T2DM, obesity, sex, and age <50 years or ≥50 years are shown in Table 2, Table 3, Table 4 and Table 5. 

For haptoglobin, in the “NAFLD-Serum” cohort, there was a cut-off effect for the lower values due to the lowest value being adjusted to 0.06 g/L. 

In the NAFLD-Biopsy subset, for subjects with T2DM, with or without obesity, all characteristics were different vs. subjects without T2DM and non-obese (controls), except for ApoA1 level and NASH prevalence. In subject with T2DM and non-obesity ApoA1 was lower. In subjects with no-T2DM and obese the advanced NASH prevalence was higher. In the NAFLD-Serum subset, all comparisons showed *p*-value < 0.001.

#### 3.1.1. A2M Normal Values

Variations of A2M were wider in men than in women, normal values of A2M were much higher before 40 years of age, and normal values of A2M increased slowly after 50 years of age.

#### 3.1.2. ApoA1 Normal Values

ApoA1 in men was very stable from 10 to 70 years of age and was much lower than in women. ApoA1 in women increased from birth to 60 years of age, with a slow decrease thereafter.

#### 3.1.3. Hapto Normal Values

Haptoglobin decreased from birth to 10 years of age, and then re-increased slowly up to 70 years of age in women and up to 50 years of age in men. 

### 3.2. Impact of T2DM on Proteins According to Histological Metabolic Liver 

#### 3.2.1. A2M Values

Type 2 diabetes was associated with a significant and earlier increase in A2M (Figure 2). In male patients with T2DM, the mean levels of A2M were significantly higher than in patients without T2DM (all *p*-values ≤ 0.05), whatever the histological confounder subset. In females with T2DM, A2M was higher than in non-T2DM but only in subsets with significant NASH or significant steatosis grades and in subsets stratified by obesity. A2M was higher than the normal values in all subsets except for women without T2DM and with stage F0F1 (n = 85), with an unexpected significant decrease in A2M versus normal values up to 65 of age, by −0.14 g/L. A2M was associated with clinically significant fibrosis (Figure 3) and NASH (Appendix A). As expected, the increase in A2M after 50 years of age was the highest in patients with significant fibrosis stage F2F3F4 (more than 0.70 g/L in men and more than 0.45 g/L in women) and significant NASH grade A2A3 (more than 0.68 g/L in men and more than 0.46 g/L in women) versus non-significant features. A2M levels were decreased in comparison with normal values in patients with significant steatosis, S2S3, before the age of 50 years and with no T2DM (Figure 4).

Regarding A2M versus normal values in women, patients with significant steatosis S2S3 started with T2DM at 50 years of age, 10 years earlier than in non-T2DM, and in men at 40 years of age. Interestingly, A2M levels were lower than normal in women before the age of 50 years, both without T2DM (−0.07 g/L) and with T2DM (−0.19 g/L). The significant increases above the normal levels started after 50 years of age, ranging from −019 to +0.38 in T2DM, and from −0.07 to +0.16 g/L in non T2DM (Figure 4). 

#### 3.2.2. ApoA1 Values

ApoA1 levels were significantly decreased in women with T2DM compared with those without T2DM from the age of 30 years, whatever the confounders. In women and men with T2DM, ApoA1 values before the age of 50 years were already significantly lower than in patients without T2DM, at −0.25 g/L and −0.09 g/L, respectively (Figure 5). T2DM was associated with lower ApoA1 (*p* < 0.001) in women with significant NASH A2A3 (Appendix A). T2DM was also associated with lower ApoA1 (*p* = 0.002) in men and women with significant steatosis S2S3 (Appendix A). Furthermore, T2DM was associated with lower ApoA1 (*p* = 0.04) in obese men (Appendix A).

#### 3.2.3. Hapto Values

T2DM was associated with a significant increase in haptoglobin levels (+0.23 g/L) versus non-T2DM (−0.11 g/L) in men (Figure 6). This association persisted in men whatever the confounders. In women, the only significant variability was an increase in haptoglobin in patients without T2DM 0.26 g/L; *p* = 0.01 (Figure 6).

### 3.3. Impact of T2DM on Serum Protein Levels According to Obesity

#### 3.3.1. A2M

A2M levels were higher in patients with T2DM versus non T2DM regardless of the presence of obesity (all *p* = 0.04). In obese women with T2DM, A2M was decreased before the age of 50 years (−0.15 g/L) and increased after 50 years of age (+0.30 g/L) compared with normal values (Figure 7). In non-obese females without T2DM, A2M was also increased after the age of 50 years, (0.25 g/L) compared with normal values. In non-obese individuals, A2M differences versus normal value ranged from 0.15 g/L in women without T2DM (Appendix A) to 0.48 g/L in obese men with T2DM (Appendix A).

#### 3.3.2. ApoA1

In non-obese (Appendix A) and obese (Appendix A) individuals, decreased ApoA1 levels were associated with T2DM in women but not in men. 

#### 3.3.3. Hapto

In non-obese (Appendix A) and obese (Appendix A) individuals, reduced haptoglobin levels were associated with T2DM (*p* = 0.003) in men but not in women. Haptoglobin levels in men, in comparison with normal values, ranged from −0.18 g/L in non-obese individuals without T2DM to +0.26 g/L in obese men with T2DM. 

#### 3.3.4. Univariate Correlations between Proteins and between Histological Features

There was no significant association between the three proteins within the four sex/obesity subsets *(*Appendix A). Among the three histological features, fibrosis stage was associated positively, as expected, with NASH grades in the four sex/obesity subsets. There was an unexpected association with steatosis grades in non-obese men. Fibrosis stage was associated with age in the four sex/obesity subsets with the exception of non-obese women (Appendix A). Both NASH and steatosis grades were not significantly associated with age. Steatosis grade was significantly and positively associated with NASH and fibrosis only in non-obese women (Appendix A). A2M level, as expected, was significantly (*p* < 0.05) and positively associated with age, fibrosis stage, and NASH grade in the four sex/obesity subsets. ApoA1 level was only associated positively with age in obese men (Appendix A). Haptoglobin was associated positively with age only in non-obese men and surprisingly positively associated with the grade of steatosis both in obese and non-obese men (Appendix A).

#### 3.3.5. Multivariate Regression

This analysis permitted adjustment of the association between T2DM and each protein, independently of the two other proteins of interest after taking into account age and the SAF class of each feature in each of the four sex/obesity subsets.

T2DM Multivariate Association with Proteins and Fibrosis (Appendix A).

A2M was not associated with the presence of T2DM in the four sex/obesity subsets. ApoA1 was significantly and negatively associated with T2DM in women with or without obesity. Haptoglobin was significantly and positively associated with T2DM in men with or without obesity.

T2DM Multivariate Association with Proteins and NASH (Appendix A).

A2M was only associated with the presence of T2DM in non-obese men. ApoA1 was significantly and negatively associated with T2DM in women with or without obesity. Haptoglobin was significantly and positively associated with T2DM in men with or without obesity.

T2DM Multivariate Association with Proteins and Steatosis (Appendix A)

A2M was not associated with the presence of T2DM in the four sex/obesity subsets. ApoA1 was significantly and negatively associated with T2DM in women without obesity. Haptoglobin was significantly and positively associated with T2DM in men without obesity.

#### 3.3.6. Impact of BMI on Serum Level Proteins

##### Impact of BMI on A2M Serum Level in the “NAFLD-Serum” Cohort

In diabetic patients, there was a higher level of A2M according to BMI vs non-diabetic patients, both in males and females, in patients above 25 kg/m^2^ (Figure 8). 

##### Impact of BMI on ApoA1 Serum Level in the “NAFLD-Serum” Cohort

In all patients, there was a negative correlation between ApoA1 and BMI (Figure 9). 

##### Impact of BMI on Haptoglobin Serum Level in the “NAFLD-Serum” Cohort

There was a strong positive correlation between Haptoglobin serum level and BMI, particularly in females (Figure 10).

### 3.4. Impact of T2DM on the Three Proteins According to SARS-CoV-2 Infection 

Before COVID-19, the means of A2M were higher and ApoA1 lower than normal expected values. The means of haptoglobin were higher than normal expected values only in obese patients (Figure 11).

During COVID-19, the means of ApoA1 were even lower than before COVID-19. The means of haptoglobin were higher than the normal expected value only in obese patients. The means of haptoglobin were even higher than before COVID-19 only in obese patients (Figure 11).

#### 3.4.1. A2M

Before COVID-19, T2DM was associated with a significant and very early increase in A2M before 30 years of age compared with non-T2DM, both in men and women (Figure 12). In women with T2DM, A2M levels were increased by 0.38 g/L compared with 0.19 g/L without T2DM, and in men by 0.76 g/L compared with 0.42 g/L without T2DM (all *p* < 0.001). During COVID-19, T2DM was associated with different A2M kinetics compared with those prior to COVID-19. The levels decreased below the normal values under the age of 20 years and increased 15 years later (45 years of age) compared with 30 years before COVID-19 (Figure 13). A2M increases were also lower during COVID-19 than before. In women with T2DM, A2M levels were increased by 28 g/L compared with 6 g/L without T2DM, and in men by 0.61 g/L compared with 0.26 g/L without T2DM (all *p* < 0.001).

#### 3.4.2. ApoA1

Before COVID-19, T2DM was associated with a significant and very early decrease in ApoA1 in patients younger than 20 years of age compared with non-T2DM patients, in both in men and women (Figure 14). In women with T2DM, ApoA1 levels were decreased by −0.18 g/L compared with −0.09 g/L in those without T2DM, and in men by −0.10 g/L compared with −0.06 g/L without T2DM (all *p* < 0.001). During COVID-19, T2DM was associated with a significant and very early decrease in ApoA1 in patients younger than 20 years of age compared with non-T2DM patients, both in men and women (Figure 15). In women with T2DM, ApoA1 levels were −0.22 g/L compared with −0.12 g/L without T2DM, and in men by −0.13 g/L compared with −0.08 g/L without T2DM (all *p* < 0.001).

#### 3.4.3. Haptoglobin

Before COVID-19, T2DM was associated with a significant and very early increase in haptoglobin in patients younger than 20 years of age compared with non-T2DM patients, both in men and women (Figure 16). In women with T2DM, ApoA1 levels were increased by 0.16 g/L compared with 0.06 g/L without T2DM, and in men by 0.05 g/L compared with 0.06 g/L without T2DM (all *p* < 0.001). During COVID-19, T2DM was associated with a significant and very early increase in haptoglobin in patients younger than 20 years of age compared with non-T2DM patients, both in women (0.25 g/L vs 0.12 g/L) and men (0.14 g/L vs 0.02 g/L) (Figure 17).

## 4. Discussion

Here, the impact of T2DM on the serum levels of three proteins in patients at risk of NAFLD was assessed according to eight major confounding factors: age, sex, obesity, NAFLD histological liver features (fibrosis, NASH steatosis), and SARS-CoV-2 infection. Such a study has never been performed before, despite the rationale and evidence base of such correlations [1,2,3,4,5,6,7,8,9,10,11,12,13,14,15,16,17,18,19,20,21,22,23,24,25,26,27,28,29,30,31,32,33,34,35,36,37,38,39,40,41,42,43,44,45,46,47,48,49,50,51,52,53,54,55,56,57,58,59,60,61].

Overall, we found that the levels of the three proteins were significantly different than normal values in patients at risk of NAFLD without T2DM, with increased A2M, decreased ApoA1, and increased haptoglobin. In patients with both NAFLD and T2DM, these significant differences were magnified. Furthermore, in cases of SARS-CoV-2 infection, this population had a third factor of decreased ApoA1 and increased haptoglobin. These results are in line with the independent diagnostic and prognostic values of ApoA1 and haptoglobin combined with A2M in COVID-19 [9,10].

In patients at risk of NAFLD, several multivariate tests used at least one of these three proteins for the non-invasive diagnosis or prognosis of liver features [73,74], as follows: A2M (FibroTest/FibroSure [35,37,75], FibroMeter V2G and V3G [73,74], Hepascore [73,74], and NIS4 [76]), ApoA1 (FibroTest/FibroSure, Chunming score, Shukla score) [35,37,77,78], and haptoglobin (FibroTest/FibroSure Fuc–Hpt–Mac2bp) [35,37,79]. 

The recent increase in the knowledge of these proteins, as well as their inclusion in multivariate biomarkers, deserves discussion regarding their clinical interest and limitations, including their respective risks of false positives and false negatives. 

### 4.1. A2M Variability in Patients at Risk of NAFLD

#### 4.1.1. Strengths

The variability of A2M according to age, in comparison with expected USA normal values, was retrieved in both the “NAFLD-Biopsy” and “NAFLD-Serum” cohorts. 

The normal values were highly different according to the four age periods, decreasing rapidly between 5 to 25 years of age, almost stable between 25 to 50, and re-increasing slowly up to more than 75 years of age. These results support both the hypothesis of a preventive role of A2M in SARS-CoV-2 infection in children up to the age of 25 years, and also the aging damage role of A2M after the age of 50 years. 

Here, A2M in T2DM was significantly increased at an earlier time point (Figure 2 and Figure 9) than in non-T2DM. This is in line not only with the role of A2M as a regulator of the extracellular matrix [19,23,26] and the well-known positive association of A2M in the extracellular matrix of liver fibrosis (Appendix A) [19,23,30,32,33,34,35,36,37,38], but also with the early damage of proteins associated with the very early presence of T2DM [19,27,28,29,80]. In men with T2DM of the “NAFLD-Serum” cohort, A2M increased as soon as the age of 40 years, reaching a mean increase of 0.88 g/L between 50 to 75 years of age (Figure 9).

#### 4.1.2. Limitations

In the “NAFLD-Biopsy” cohort, the sample size was too small to assess the kinetics before the age of 25 years, and therefore to assess the association with liver features. However, thanks to the large sample size of the “NAFLD-Serum” cohort, the sample size of patients at risk before the age of 25 years was sufficient to compare curves of A2M in patients with T2DM2 vs. normal expected values, at least in USA NAFLD patients. Here, thanks to the large sample size, we observed a decrease in the mean A2M in obese patients at risk of NAFLD, with or without T2DM, when compared with normal USA values adjusted for age and sex, as described in other contexts of use. 

A2M is a candidate biomarker mirroring key metabolic steps for health and disease. It gains insight into the interaction of anabolism with catabolism [81]. The results of several studies in Thai adults in Bangkok showed different associations between A2M serum levels compared with normal Thai individuals, with lower levels than in the USA. In hard-working male construction laborers, a negative correlation was found for the variables age, weight, height, BMI, and HDL, with A2M as the dependent variable [82]. A2M of female construction workers did relate to any of the variables investigated. A dietary survey conducted with apparently health Thai farmers found a statistically significant negative correlation of A2M with energy, protein, fat, and carbohydrate intake [83]. Tobacco smoking could be a confounding factor [84,85]. All these results obtained from the variety of different studies seem indeed to be in accordance with the assumption that A2M supports homeostasis in situations of a “challenged” nutritional status [83].

In the “NAFLD-Biopsy” cohort, A2M levels were decreased in comparison with normal values in patients with significant steatosis S2S3 before the age of 50 years and without T2DM (Figure 4). Such an unexpected result requires confirmation and further physio–pathological evidence. Degradations of the proteins could be an explanation [86]. 

During SARS-CoV-2 studies in non-human primates, we recently observed an unexpected very early decrease in serum A2M 2 days post-infection at the peak of nasopharyngeal viral loads, with a return to baseline values at the seventh day [10]. Similar kinetics of A2M were observed in hospitalized patients with COVID-19 not requiring intensive care [9,10]. If confirmed, this kinetic could be associated with the rapid consumption or degradation of native A2M during the peak of the acute phase response. Similar early kinetics of A2M levels were observed in hemodialysis patients, with lower levels during COVID-19 vs. healthy controls, and vs. hemodialysis patients without COVID-19 [11].

#### 4.1.3. Causal Relationships with Clinical Endpoints

In patients with T2DM, the predominant circulating form of A2M is degraded [86]. In nephrotic syndrome, serum A2M levels start to rise when a trace amount of albumin is excreted. Hepatic synthesis of A2M is enhanced significantly to replace lost liver-derived proteins in experimental animals and humans, resulting in a net increase in its serum levels [87,88,89]. Because of this risk of the confounding factors, we checked the absence of correlations (logistic regression) between the presence of obesity and kidney function assessed by CKD index and A2M levels adjusted by age and sex in the T2DM cohort post–hoc. There was no significant association between A2M and kidney function (Appendix A).

In the plasma proteomic profile of T2DM patients submitted to bariatric surgery, A2M significantly increased in individuals whose diabetes persisted or remitted after weight loss, using non-diabetic but similarly obese persons as controls [90]. Therefore, in this context, A2M could be a prognostic marker. 

### 4.2. ApoA1

#### 4.2.1. Strengths

Our epidemiological results support the causal relationship between low ApoA1 as an independent risk factor of infection by SARS-CoV-2 in NAFLD patients with additive risks of T2DM and obesity [2,3,4,5,6,7,8,9,10,13,91]. 

In the “NAFLD-Biopsy” cohort, ApoA1 levels were already significantly decreased (−0.30 g/L) in women with T2DM compared with non-T2DM patients from the age of 30 years (Figure 5), whatever the confounders (Supplementary Appendix A). This early decrease in patients with biopsy was retrieved in the “NAFLD-Serum” cohort with a mean decrease of 0.20 g/L (Figure 5). In men, the ApoA1 decrease compared with normal values was also significant, but much less than in women where the levels increased significantly from birth to 60 years of age. 

The association of T2DM with lower ApoA1 (*p* < 0.001) in women with histologically significant NASH, grades A2A3, is original and in line with other more severe liver diseases with high grades of necro-inflammatory inflammation, such as severe alcoholic hepatitis, which was described in previous studies. 

We used the “NAFLD-Serum” cohort to confirm the previous findings [9]: that before (Figure 14) and during (Figure 15) COVID-19, T2DM was associated with a decrease in ApoA1 compared with non-T2DM patients, in both in men and women. For the first time, to the best of our knowledge, we observed that this decrease in ApoA1 was lower in obese vs. non-obese, both before and during COVID-19 (Figure 11). 

#### 4.2.2. Limitations

The present results were limited by the absence of adjustments on confounding factors associated with serum ApoA1 levels, such as dietary folate, physical exercise, and vitamin C [92] and alcohol intake both before [93] and during [94] COVID-19. 

#### 4.2.3. Causal Relationships with Clinical Endpoints

Recent phase 2 randomized trials in patients with NAFLD used serum ApoA1 as secondary endpoints for assessing the drug’s benefit [68,95]. 

Lanifibranor is a pan-PPAR (peroxisome proliferator-activated receptor) agonist that modulates key metabolic, inflammatory, and fibrogenic pathways in the pathogenesis of NASH [68]. 

Pegbelfermin (PGBF) is a hormone that can reduce bile acids (BA) that have previously been shown to have toxic effects on the liver. 7a-hydroxy-4-cholesten-3-one (C4), a biomarker of primary BA synthesis in the liver, was measured in plasma of the patients with NASH together with ApoA1. After PGBF treatment, C4 and ApoA1 were increased vs. control, and associated with decreases in the LDL/HDL ratio, suggesting that PGBF-associated changes in BA metabolism may contribute to an improvement in lipoprotein profiles that could possibly lead to a reduction in cardiovascular risk in patients with NASH [96,97].

Numerous experimental results explained how ApoA1 may lose its functionality in many inflammatory and pathological conditions, including T2DM, liver diseases, obesity and COVID-19 [41]. ApoA1 is secreted by the liver (about 70%) and the intestine (about 30%). This lipid free ApoA1 interacts with the ATP-binding cassette transporter A1 (ABCA1) on peripheral cells, leading to the transfer of cholesterol and cellular phospholipids from the cell membrane to ApoA1. T2DM, liver diseases, obesity and COVID-19 can impair the biogenesis of ApoA1, as well as its post-translational modifications, including oxidation, carbamylating, and glycation.

Several recent studies separating the roles of the main HDL components, such as HDL2, HDL3, and ApoA1 (or according to the HDL size), have suggested that the causal relationships between serum levels of ApoA1 with clinical endpoints such as cardiovascular events [13], risk of severe COVID-19 [14], or glycosylated hemoglobin [96] were easier to prove than using the HDL overall levels.

Finally, a recent work demonstrating a new source of ApoA1 for the liver through the portal vein directly from the intestine opens up many possible causal mechanisms with the risks of T2DM, liver diseases, obesity, and COVID-19 [53]. HDL3 produced by the intestine protects the liver from the inflammation and fibrosis observed in a variety of mouse models of liver injury that parallel clinically relevant conditions in humans, including surgical resection of the small bowel, alcohol consumption, or high-fat diets. 

### 4.3. Haptoglobin

#### 4.3.1. Strengths

The present results underline the performance of haptoglobin as a sensitive biomarker of inflammation in chronic liver disease. Thus far, the clinical utility of serum haptoglobin is its prognostic value in multivariate blood tests when its level is decreased, which has been observed mostly in advanced fibrosis stages.

Here, in patients at risk of NAFLD, haptoglobin levels were positively associated with male sex, T2DM, and obesity, and surprisingly with the grade of steatosis in non–obese men, in univariate and multivariate analyses. These associations are in line with a chronic inflammation profile in these patients before the onset of histological NASH [54,55,56,57,58,59,60,61].

During SARS-CoV-2 studies in non-human primates, we recently observed that haptoglobin levels, in comparison with CRP, had the same early increase 2 days after infection, but remained more consistently elevated for at least 10 days post-infection [10]. Recent human studies underlined the clinical interest in haptoglobin in SARS-CoV-2 infection [16,17,18], including one that described the normal distribution of haptoglobin versus the bimodal distribution of CRP [16].

Finally, for the first time, in patients at risk of NAFLD, the correlation between haptoglobin serum level and BMI was demonstrated, probably due to a chronic inflammation. 

#### 4.3.2. Limitations

The present results were limited by the absence of adjustments on confounding factors associated with serum haptoglobin levels, such as haptoglobin polymorphism, inflammatory bowel disease, hemolysis, iron deficiency, and exercise [98,99,100]. Several studies found associations between haptoglobin 2-2 and liver fibrosis, but the causality has not been validated thus far [16].

#### 4.3.3. Causal Relationships with Clinical Endpoints

Haptoglobin increase is a validated direct early consequence of the acute phase reaction. However, as for ApoA1, serum haptoglobin variability observed in T2DM, obesity, and COVID-19 could also be directly due to the intestine, such as changes in endothelial permeability. Higher serum levels of haptoglobin were observed in obese patients with increased jejunal permeability revealed by lipid challenge and linked to inflammation and T2DM [101]. After a lipid load, haptoglobin was two-fold higher in obese patients compared to non-obese controls and correlated with systemic and intestinal inflammation. Lipid-induced permeability was associated with the presence of T2DM and obesity. In such correlations, the mechanisms explaining the variability of serum haptoglobin could be both genetic defects (such as Gata6 gene and a decrease in zonulin—the pre-haptoglobin protein), and specific environmental factors (such as high-fat diet, alcohol, fiber-deprived diet, bacterial or viral infection, and medication exposure) known to contribute to break the intestinal barrier balance and promote gut dysbiosis [102]. 

In mice, exercise before a polyunsaturated fatty acid-based (PUFA) diets upregulates haptoglobin fourfold and eightfold according to isocaloric or ad libitum diets, respectively [103]. In both humans and mice, haptoglobin has been shown to increase in plasma because of increased secretion both by the liver and adipose tissue, and is increased by many factors including tumor necrosis factor–alpha and interleukin–6. These cytokines are known to be increased in their abundance when animals are placed on a high PUFA diet [104]. A consequence of a haptoglobin increase is increased plasma abundance on an HFD, which may include increased risk for cardiovascular disease [105].

### 4.4. Methodological Limitations

The main limitations of the present study are those of the epidemiological study in the non-interventional cohort, which identified significant associations but with multiple tests, many confounding factors, several risks of collinearity, and few direct proofs of causality. 

Furthermore, the impact of T2DM on the three proteins according to SARS-CoV-2 infection was indirectly estimated in a large population at risk of NAFLD, but without direct virological markers. We retrieved the same kinetics of decreased ApoA1 during the successive waves of SARS-CoV-2 infection in comparison with those in 2019, but new confounders appeared such as decreased physical exercise and increased tobacco and/or alcohol consumption [93,94]. However, concerning ApoA1, an increase in alcohol consumption would have been associated with an increase in ApoA1 [50,51] in the “NAFLD-Serum” cohort, which included 78% of subjects without advanced fibrosis [9]. The body weight means were also similar (88.5 kg) in 2019 and 2020 [9]. 

In epidemiological studies of patients with T2DM, several variable and bias factors remained possible candidates, such as the definition of T2DM (clinical definition by diabetologists or fasting glucose) and the number of treatments of protein levels [69]. In the meantime, these variable factors should be discussed in the context of the use of such blood test components. Here, the results of the “Control Population” were derived from a relatively homogeneous Caucasian population, and the findings may not be applicable to other ethnic groups. The “NAFLD-Biopsy” cohort was limited by the selection of patients who accepted a liver biopsy in tertiary centers and their enrollment according to abnormal ALT or the presence of steatosis at ultrasonography. The “NAFLD-Serum” cohort was limited by the absence of ethnic origin information and the few available clinical characteristics, including age, sex, BMI, ApoA1, A2M, haptoglobin, liver function tests, fasting glucose, total cholesterol, and triglycerides. 

Finally, forty years ago, clinicians were not using A2M, ApoA1 was interesting for predicting cardiovascular diseases, and haptoglobin was mostly used for the diagnosis of hemolysis. Nowadays, these proteins are widely prescribed in multivariate noninvasive tests for the diagnosis of liver diseases. However, these three proteins—without collinearity between them but with ubiquitous functions now better understood—should permit the construction of better multivariate tests in metabolic diseases for cumulating the risk of liver diseases and the risk of severe infections such as T2DM, NAFLD, and COVID-19. 

## Figures and Tables

**Figure 1 biomedicines-10-00699-f001:**
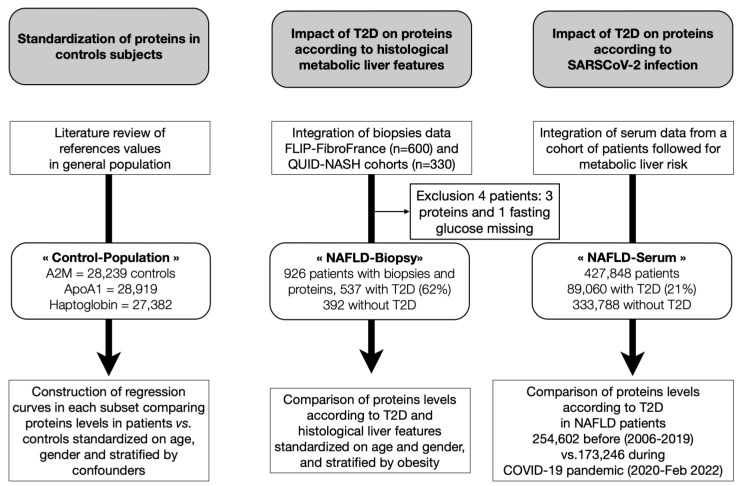
Study design.

**Figure 2 biomedicines-10-00699-f002:**
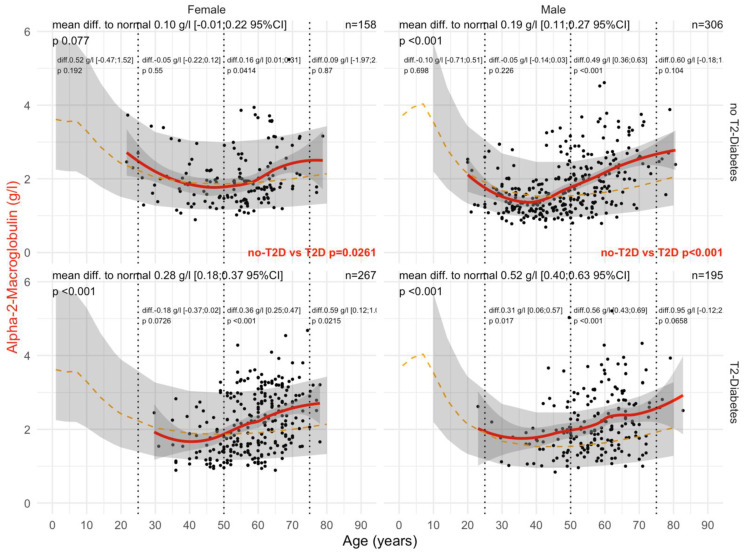
A2M levels in all “NAFLD-Biopsy” patients. Impact of T2DM on A2M concentration in all “NAFLD-Biopsy” patients (n = 926). Normal serum mean values according to age and sex (dashed–orange lines), with 95% confidence intervals (light-gray ribbon). Three vertical dotted lines mark the four age groups (before 25 years, between 25 and 50 years, between 50 and 75 years, and above 75 years old). Each point is a patient’s protein value. The red curve is a Loess regression of the median A2M value, along with its 95% confidence interval (darker gray). For each age group, the mean difference (%95CI) between the patient protein value and the expected normal value (for age and sex) with its significant *p*-value is displayed at the top of the figure. The significance between the non-T2DM and T2DM patients is displayed in red between the two panels.

**Figure 3 biomedicines-10-00699-f003:**
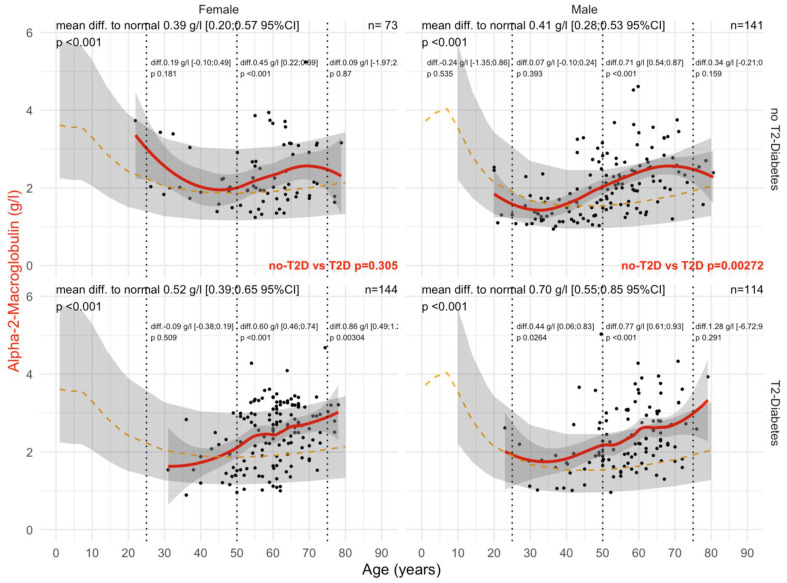
A2M levels in “NAFLD-Biopsy” patients with significant fibrosis stages F2F3F4. Impact of T2DM on A2M concentration in NAFLD-biopsied patients with clinically significant fibrosis F2F3F4 (n = 472). Normal serum means values according to age and sex (dashed–orange lines), with 95% confidence intervals (light-gray ribbon). Three vertical–dotted lines mark the four age groups (before 25, between 25 and 50, between 50 and 75, and above 75 years old). Each point is a patient’s protein value. The red curve is a Loess regression of the median A2M values, along with its 95% confidence interval (darker gray). For each age group, the mean difference (%95CI) between the patient protein value and the expected normal value (for age and sex) with its significance *p*-value is displayed at the top of the figure. The significance between the non-T2DM and T2DM patients is displayed in red between the 2 panels.

**Figure 4 biomedicines-10-00699-f004:**
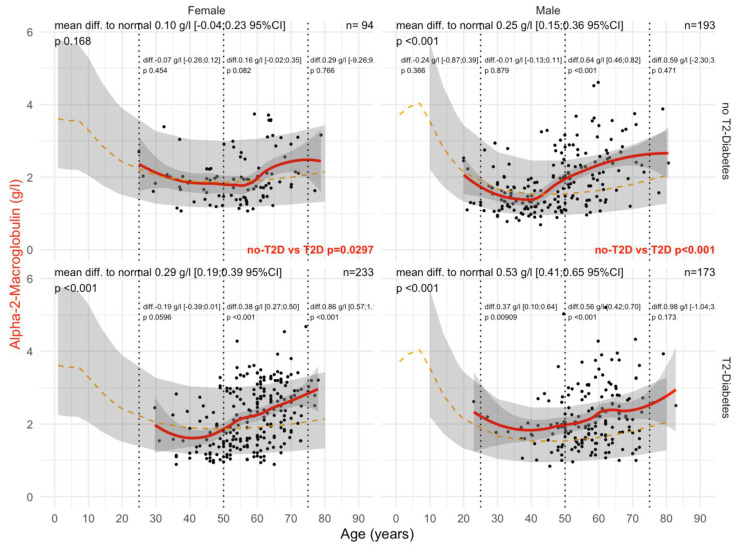
A2M levels in “NAFLD-Biopsy” patients with significant steatosis stages S2S3. Impact of T2DM on A2M concentration in NAFD-biopsied patients with significant steatosis S2S3 (n = 693). Normal serum means values according to age and sex (dashed–orange lines), with 95% confidence intervals (light-gray ribbon). Three vertical–dotted lines mark the four age groups (before 25, between 25 and 50, between 50 and 75, and above 75 years old). Each point is a patient’s protein value. The red curve is a Loess regression of the median of A2M values, along with its 95% confidence interval (darker gray). For each age group, the mean difference (%95CI) between patient protein value and the expected normal value (for age and sex) with its significance *p*-value is displayed at the top of the figure. The significance between the non-T2DM and T2DM patients is displayed in red between the 2 panels.

**Figure 5 biomedicines-10-00699-f005:**
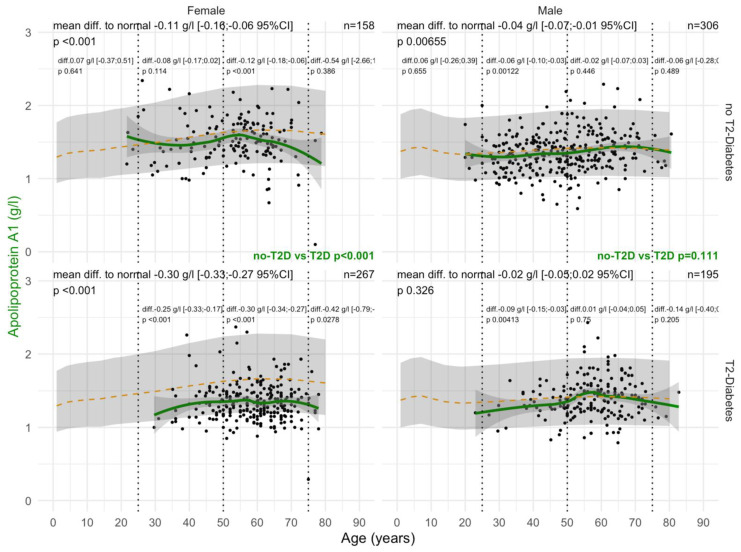
ApoA1 levels in all “NAFLD-Biopsy” patients. Impact of T2DM on ApoA1 concentration in all NAFD-biopsied patients (n = 926). Normal serum means according to age and sex (dashed–orange lines), with 95% confidence intervals (light-gray ribbon). Three vertical–dotted lines mark the four age groups (before 25, between 25 and 50, between 50 and 75, and above 75 years old). Each point is a patient’s protein value. The green curve is a Loess regression of the median of ApoA1 values, along with its 95% confidence interval (darker gray). For each age group, the mean difference (%95CI) between the patient protein value and the expected normal value (for age and sex) with its significance *p*–value is displayed at the top of the figure. The significance between the non-T2DM and T2DM patients is displayed in green between the 2 panels.

**Figure 6 biomedicines-10-00699-f006:**
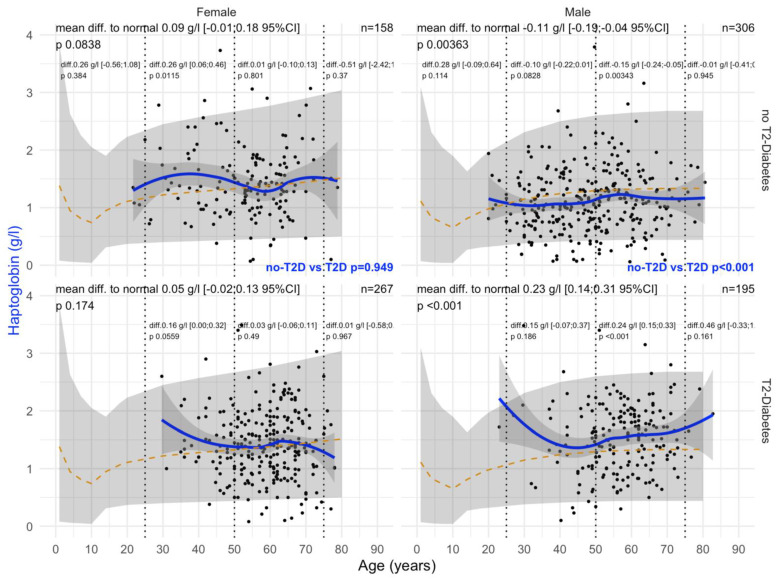
Haptoglobin levels in all “NAFD-Biopsy” patients. Impact of T2DM on Hapto concentration in all NAFD-biopsied patients (n = 926). Normal serum means values according to age and sex (dashed–orange lines), with 95% confidence intervals (light-gray ribbon). Three vertical–dotted lines mark the four age groups (before 25, between 25 and 50, between 50 and 75, and above 75 years old). Each point is a patient’s protein value. The green curve is a Loess regression of the median of Hapto values, along with its 95% confidence interval (darker gray). For each age group, the mean difference (%95CI) between the patient protein value and the expected normal value (for age and sex) with its significance *p*-value is displayed at the top of the figure. The significance between the non-T2DM and T2DM patients is displayed in blue between the 2 panels.

**Figure 7 biomedicines-10-00699-f007:**
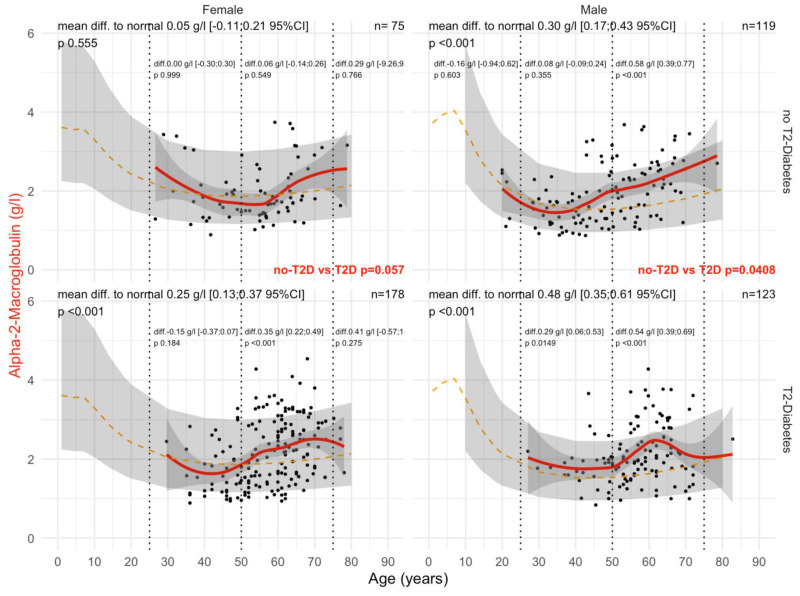
A2M levels in “NAFLD-Biopsy” patients with obesity. Impact of T2DM on A2M concentration in NAFD-biopsied patients according to obesity (n = 495). Normal serum means values according to age and sex (dashed–orange lines), with 95% confidence intervals (light-gray ribbon). Three vertical–dotted lines mark the four age groups (before 25, between 25 and 50, between 50 and 75, and above 75 years old). Each point is a patient’s protein value. The red curve is a Loess regression of the median of A2M values, along with its 95% confidence interval (darker gray). For each age group, the mean difference (%95CI) between the patient protein value and the expected normal value (for age and sex) with its significance *p*-value is displayed at the top of the figure. The significance between the non-T2DM and T2DM patients is displayed in red between the 2 panels.

**Figure 8 biomedicines-10-00699-f008:**
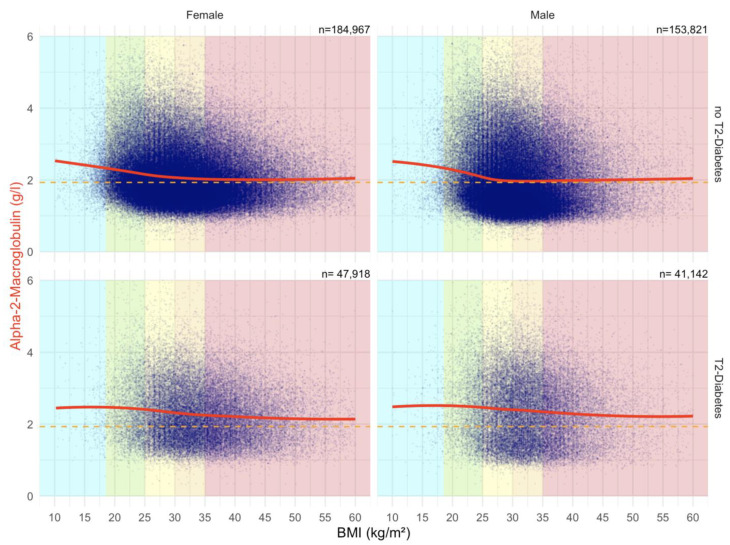
Impact of BMI on the A2M concentrations in “NAFD–Serum” cohort. Median value is the horizontal orange–dashed line. Five vertical zones mark the 5 BMI groups (WHO definition cut-offs: 18.5, 25, 30, 35 and above 40 kg/m^2^). Each blue point is a protein patient value with its BMI. The red curve is a Loess regression of the median of A2M values.

**Figure 9 biomedicines-10-00699-f009:**
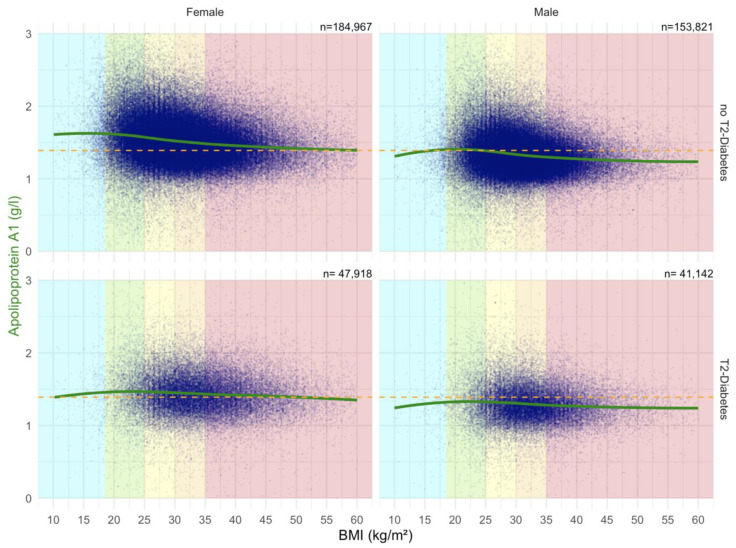
Impact of BMI on the ApoA1 concentrations in “NAFD–Serum” cohort. Median value is the horizontal orange–dashed line. Five vertical zones mark the 5 BMI groups (WHO definition cut-offs: 18.5, 25, 30, 35 and above 40 kg/m^2^). Each blue point is a protein patient value with its BMI. The green curve is a Loess regression of the median of ApoA1 values.

**Figure 10 biomedicines-10-00699-f010:**
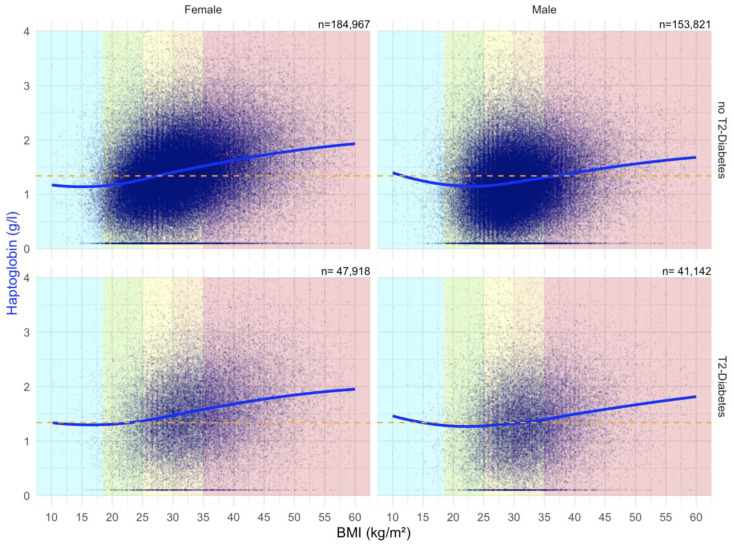
Impact of BMI on the Haptoglobin concentrations in “NAFD–Serum” cohort. Median value is the horizontal orange–dashed line. Five vertical zones mark the 5 BMI groups (WHO definition cut-offs: 18.5, 25, 30, 35 and above 40 kg/m^2^). Each blue point is a protein patient value with its BMI. The blue curve is a Loess regression of the median of Haptoglobin values.

**Figure 11 biomedicines-10-00699-f011:**
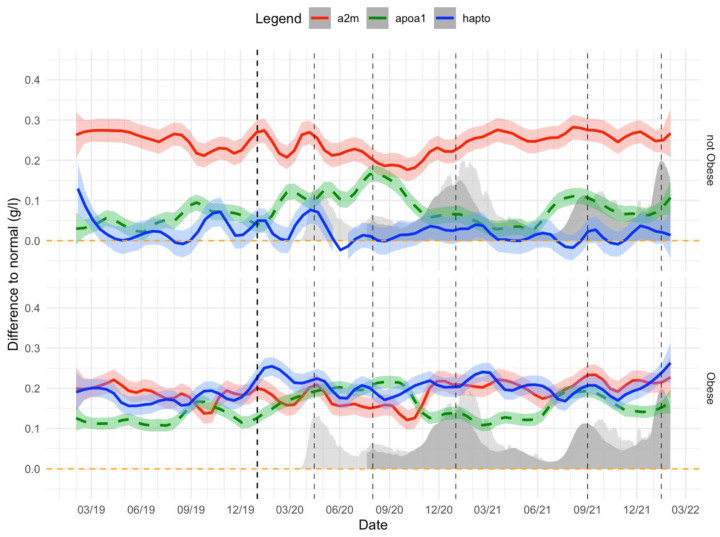
A2M, ApoA1, and haptoglobin levels before and during the COVID-19 pandemic. Impact of COVID-19 waves on A2M, ApoA1 and haptoglobin level differences with normal values before and during COVID-19 pandemic in obese and non-obese “NAFLD-Serum” subjects followed for metabolic liver risk (n = 135,911). *X*-axis is time, between January 2019 and February 2022. Dashed–vertical black lines show the different COVID-19 waves. In light gray and dark gray are the standardized COVID-19-related death and hospitalization rates, respectively, in the USA. The red line, blue line, and dashed–green line are the 7-day rolling mean difference and %95CI between the observed A2M, haptoglobin, and negative Apoa1 concentration, respectively, and the expected normal values for these subjects adjusted by their age and gender.

**Figure 12 biomedicines-10-00699-f012:**
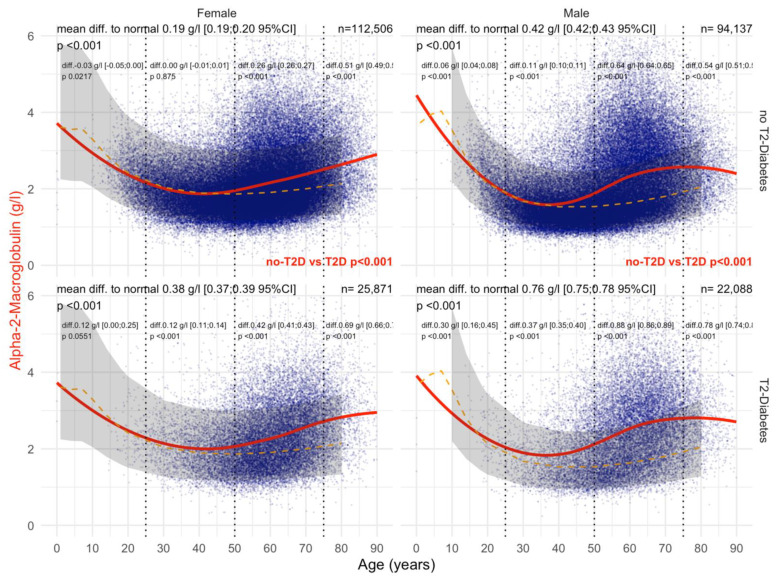
A2M levels before the COVID-19 pandemic. Impact of T2DM on A2M concentration before SARS-CoV-2 pandemic in “NAFLD-Serum” patients followed for metabolic liver risk (n = 254,602). Normal serum mean values according to age and sex (dashed–orange lines), with 95% confidence intervals (gray ribbon). Three vertical–dotted lines mark the four age groups (before 25, between 25 and 50, between 50 and 75, and above 75 years old). The red curve is a Loess regression of the median of A2M values. For each age group, the mean difference (%95CI) between the patient protein value and the expected normal value (for age and sex) with its significance *p*-value is displayed at the top of the figure. The significance between the non-T2DM and T2DM patients is displayed in red between the 2 panels.

**Figure 13 biomedicines-10-00699-f013:**
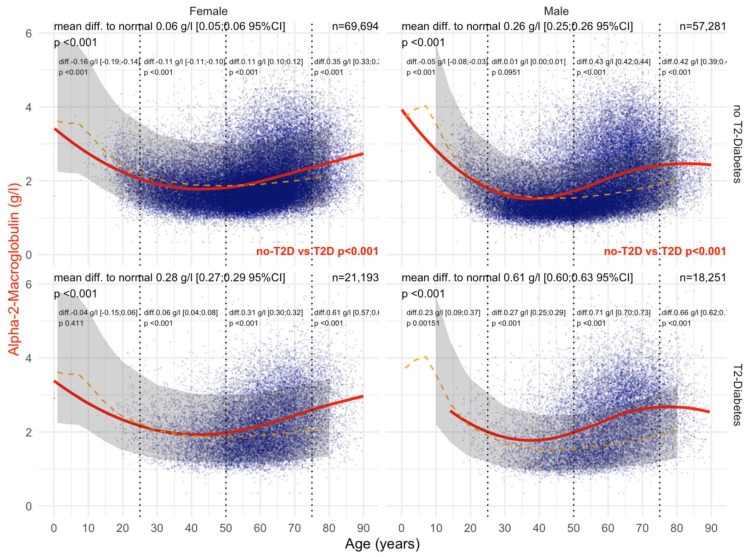
A2M levels during the COVID-19 pandemic. Impact of T2DM on A2M concentration during SARS-CoV-2 pandemic in “NAFLD-Serum” patients followed for metabolic liver risk (n = 173,246). Normal serum means values according to age and sex (dashed–orange lines), with 95% confidence intervals (gray ribbon). Three vertical–dotted lines mark the four age groups (before 25, between 25 and 50, between 50 and 75, and above 75 years old). The red curve is a Loess regression of the median of A2M values. For each age group, the mean difference (%95CI) between the patient protein value and the expected normal (for age and sex) with its significance *p*-value is displayed at the top of the figure. The significance between the non-T2DM and T2DM patients is displayed in red between the 2 panels.

**Figure 14 biomedicines-10-00699-f014:**
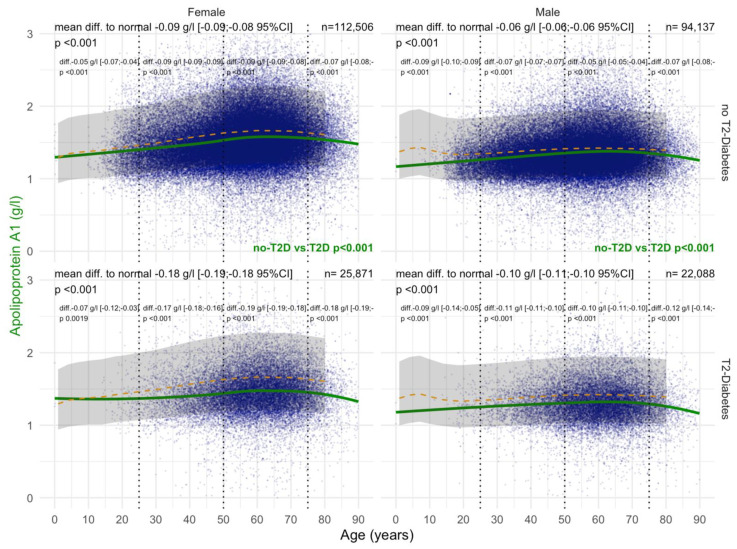
ApoA1 levels before the COVID-19 pandemic. Impact of T2DM on ApoA1 concentration before SARS-CoV-2 pandemic in “NAFLD-Serum” patients followed for metabolic liver risk (n = 254,602). Normal serum means values according to age and sex (dashed–orange lines), with 95% confidence intervals (gray ribbon). Three vertical–dotted lines mark the four age groups (before 25, between 25 and 50, between 50 and 75 and above 75 years old). The green curve is a Loess regression of the median of ApoA1 values. For each age group, the mean difference (%95CI) between the patient protein value and the expected normal value (for age and sex) with its significance *p*-value is displayed at the top of the figure. The significance between the non-T2DM and T2DM patients is displayed in green between the 2 panels.

**Figure 15 biomedicines-10-00699-f015:**
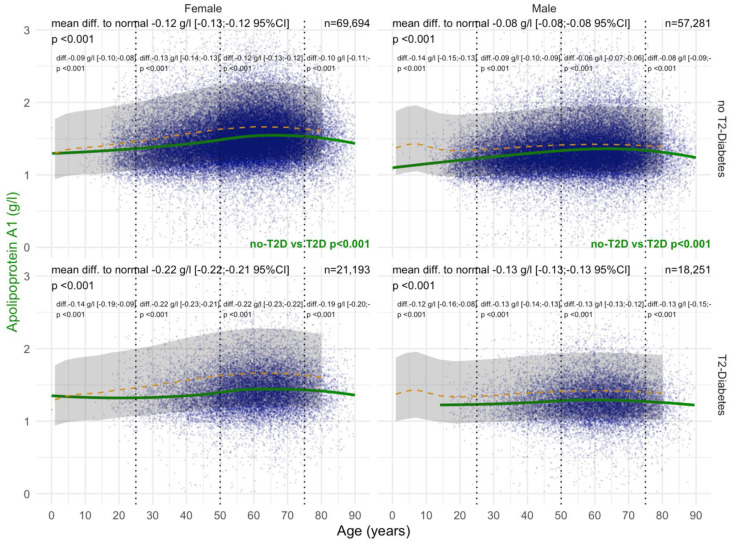
ApoA1 levels during the COVID-19 pandemic. Impact of T2DM on ApoA1 concentration during SARS-CoV-2 pandemic in “NAFLD-Serum” patients followed for metabolic liver risk (n = 173,246). Normal serum means values according to age and sex (dashed–orange lines), with 95% confidence intervals (gray ribbon). Three vertical–dotted lines mark the four age groups (before 25, between 25 and 50, between 50 and 75, and above 75 years old). The green curve is a Loess regression of the median of ApoA1 values. For each age group, the mean difference (%95CI) between the patient protein value and the expected normal value (for age and sex) with its significance *p*-value is displayed on top of figure. The significance between the non-T2DM and T2DM patients is displayed in green between the 2 panels.

**Figure 16 biomedicines-10-00699-f016:**
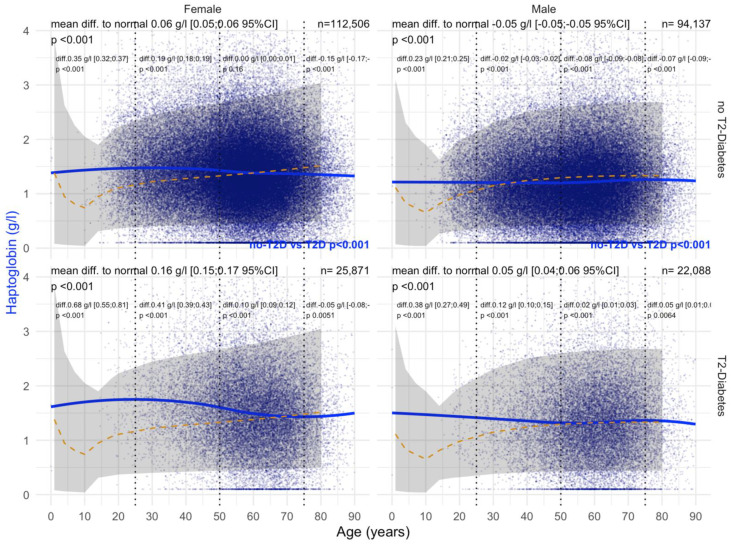
Haptoglobin levels before the COVID-19 pandemic. Impact of T2DM on Hapto concentration before SARS-CoV-2 pandemic in “NAFLD-Serum” patients followed for metabolic liver risk (n = 254,602). Normal serum means values according to age and sex (dashed–orange lines), with 95% confidence intervals (gray ribbon). Three vertical–dotted lines mark the four age groups (before 25, between 25 and 50, between 50 and 75, and above 75 years old). The blue curve is a Loess regression of the median of Hapto values. For each age group, the mean difference (%95CI) between the patient protein value and the expected normal value (for age and sex) with its significance *p*-value is displayed at the top of the figure. The significance between the non-T2DM and T2DM patients is displayed in blue between the 2 panels.

**Figure 17 biomedicines-10-00699-f017:**
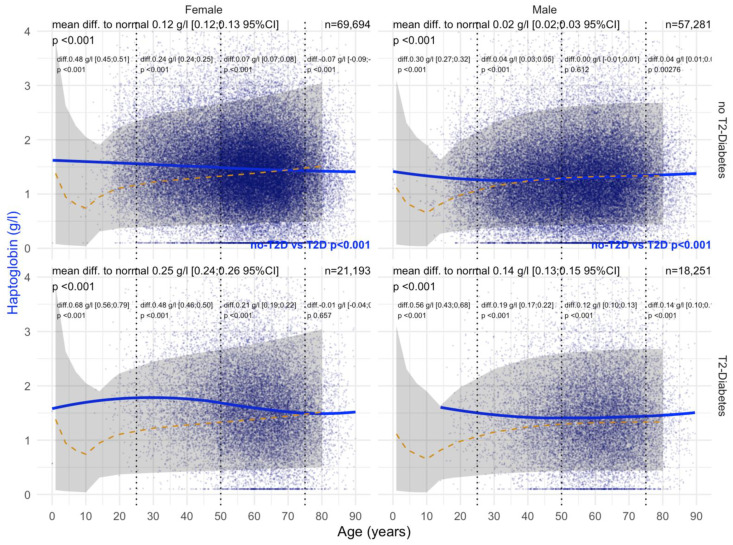
Haptoglobin levels during the COVID-19 pandemic. Impact of T2DM on Hapto concentration during SARS-CoV-2 pandemic in “NAFLD-Serum” patients followed for metabolic liver risk (n = 173,246). Normal serum means values according to age and sex (dashed–orange lines), with 95% confidence intervals (gray ribbon). Three vertical–dotted lines mark the four age groups (before 25, between 25 and 50, between 50 and 75, and above 75 years old). The blue curve is a Loess regression of the median of Hapto values. For each age group, the mean difference (%95CI) between the patient protein value and the expected normal value (for age and sex) with its significance *p*-value is displayed at the top of the figure. The significance between the non-T2DM and T2DM patients is displayed in blue between the 2 panels.

**Table 1 biomedicines-10-00699-t001:** Characteristics of “NAFLD-Biopsy” and NAFLD subsets according to diabetes and obesity. * = Significant vs. No-T2DM and Non-Obese subset.

	No-T2DM and Non-Obese	T2DM and Non-Obese	No-T2DM and Obese	T2DM and Obese	All
NAFLD-Biopsy					
N	270	161	194	301	926
Male %	187 (69.3%)	72 (44.7%) *	119 (61.3%)	123 (40.9%) *	501 (54.1%)
Age year (IQR)	49.9 (39.8–58.9)	60.0 (52.5–67.0) *	51.5 (40.7–60.8)	59.0 (51.0–64.0) *	55 (46–63) *
A2M g/L	1.67 (1.29–2.13)	2.16 (1.53–2.83) *	1.79 (1.34–3.07)	2.03 (1.56–2.74) *	1.86 (1.39–2.57)
ApoA1 g/L	1.41 (1.23–1.61)	1.33 (1.17–1.50) *	1.37 (1.21–1.55)	1.34 (1.22–1.49)	1.37 (1.21–1.55)
Hapto g/L	1.13 (0.80–1.46)	1.35 (1.00–1.76) *	1.30 (0.86–1.68) *	1.49 (1.11–1.92) *	1.31 (0.92–2.12)
Advanced fibrosis	98 (36.3%)	85 (52.8%) *	116 (59.8%) *	173 (57.5%) *	472 (51.0%)
Advanced NASH	184 (68.2%)	93 (57.8%)	157 (80.9%) *	205 (68.1%)	639 (69.0%)
Advanced steatosis	153 (56.7%)	130 (90.8%) *	134 (69.1%) *	276 (91.7%) *	693 (74.8%)
NAFLD-Serum					
N	160,136	29,439	178,652	59,621	427,848
Male %	75,519 (47.2)	14,779 (50.2)	78,302 (43.8)	26,363 (44.2)	194,963 (45.6)
Age year (IQR)	57.4 (46.2–66.4)	62.8 (54.7–70.3)	54.3 (43.4–63.3)	58.9 (50.5–66.9)	56.8 (46.2–65.6)
A2M g/L	1.91 (1.52–2.53)	2.29 (1.69–3.02)	1.83 (1.44–2.42)	2.17 (1.63–2.81)	1.93 (1.50–2.57)
ApoA1 g/L	1.44 (1.25–1.68)	1.37 (1.19–1.58)	1.37 (1.21–1.57)	1.34 (1.18–1.52)	1.39 (1.22–1.60)
Hapto g/L	1.19 (0.82–1.59)	1.31 (0.87–1.76)	1.42 (1.01–1.85)	1.51 (1.06–1.99)	1.34 (0.92–1.78)

**Table 2 biomedicines-10-00699-t002:** Characteristics and median protein levels in women <50 years of age according to T2DM and obese status.

	No-T2DM and Non-Obese	T2DM and Non-Obese	No-T2DM and Obese	T2DM and Obese
N	23,007	1983	35,303	7891
Age year (IQR)	40.1 (32.6–45.8)	43.7 (37.9–47.4)	40.3 (33.1–45.8)	43.11 (37.3–47.2)
A2M g/L	1.89 (1.57–2.29)	1.95 (1.55–2.42)	1.76 (1.46–2.16)	1.92 (1.53–2.37)
ApoA1 g/L	1.47 (1.29–1.69)	1.39 (1.21–1.60)	1.38 (1.23–1.57)	1.35 (1.20–1.53)
Hapto g/L	1.21 (0.86–1.58)	1.43 (0.99–1.86)	1.61 (1.22–2.03)	1.74 (1.31–2.22)

**Table 3 biomedicines-10-00699-t003:** Characteristics and median protein levels in women ≥50 years of age according to T2DM and obese status.

	No-T2DM and Non-Obese	T2DM and Non-Obese	No-T2DM and Obese	T2DM and Obese
N	61,610	12,677	65,047	25,367
Age year (IQR)	63.0 (57.0–69.9)	65.3 (58.9–71.8)	61.3 (55.8–67.6)	62.5 (56.7–68.9)
A2M g/L	2.02 (1.65–2.63)	2.32 (1.76–3.01)	1.95 (1.57–2.54)	2.20 (1.70–2.81)
ApoA1 g/L	1.57 (1.37–1.80)	1.45 (1.27–1.65)	1.49 (1.32–1.68)	1.43 (1.26–1.61)
Hapto g/L	1.27 (0.89–1.66)	1.35 (0.93–1.80)	1.49 (1.08–1.92)	1.57 (1.11–2.05)

**Table 4 biomedicines-10-00699-t004:** Characteristics and median protein levels in men <50 years of age according to T2DM and obese status.

	No-T2DM and Non-Obese	T2DM and Non-Obese	No-T2DM and Obese	T2DM and Obese
N	27,744	2467	33,759	6303
Age year (IQR)	39.3 (32.4–45.0)	43.5 (38.1–47.1)	39.7 (32.7–45.3)	43.3 (37.9–47.2)
A2M g/L	1.53 (1.26–1.96)	1.69 (1.29–2.35)	1.47 (1.20–1.92)	1.70 (1.28–2.29)
ApoA1 g/L	1.31 (1.16–1.48)	1.27 (1.12–1.46)	1.25 (1.12–1.40)	1.25 (1.11–1.40)
Hapto g/L	1.06 (0.72–1.44)	1.22 (0.80–1.65)	1.28 (0.91–1.68)	1.40 (0.98–1.87)

**Table 5 biomedicines-10-00699-t005:** Characteristics and median protein levels in men ≥50 years of age according to T2DM and obese status.

	No-T2DM and Non-Obese	T2DM and Non-Obese	No-T2DM and Obese	T2DM and Obese
N	47,775	12,312	44,543	20,006
Age year (IQR)	62.9 (56.9–69.5)	64.8 (58.4–71.2)	60.9 (55.5–67.3)	62.5 (56.8–68.6)
A2M g/L	2.04 (1.52–2.88)	2.48 (1.77–3.20)	2.07 (1.49–2.82)	2.44 (1.76–3.09)
ApoA1 g/L	1.36 (1.19–1.57)	1.30 (1.13–1.49)	1.30 (1.15–1.47)	1.27 (1.13–1.43)
Hapto g/L	1.16 (0.77–1.59)	1.26 (0.81–1.73)	1.29 (0.88–1.71)	1.38 (0.95–1.85)

## Data Availability

Almost all data and all results were available in the manuscript and in Appendix A. Contact olivier@biopredictive.com for more information.

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
