# Peer review of "Clinical Interest of Serum Alpha-2 Macroglobulin, Apolipoprotein A1, and Haptoglobin in Patients with Non-Alcoholic Fatty Liver Disease, with and without Type 2 Diabetes, before or during COVID-19"

_biomedicines, 2022, doi:10.3390/biomedicines10030699_

Round 1

Reviewer 1 Report

The authors conduct an observational study and aimed to investigate the effect of alpha-2 macroglobulin, apolipoprotein A1 and haptoglobin, in patients with non-alcoholic fatty 3 liver disease, with and without type-2 diabetes, before or during Covid-19. Three datasets were used: "Control-Population"(N=27,382), “NAFLD-Biopsy” cohort (N=926); and USA “NAFLD-Serum” cohort for protein kinetics before and during COVID-19 (N=421,021).

Comments:

1.

Abstract: NAFLD-Serum, n=421,021

Table 1: NAFLD-Serum, n= 427,848

It is a different sample sizes.

2.

In Table 1, characteristics of NAFLD-Biopsy and NAFLD subsets was category according to T2DM and Obese

In figures 2, 5 and 6, characteristics of NAFLD-Biopsy was category according to T2DM and sex.

It is a different information (need to same category?).

Similarly, in Figures 3 and 4, fibrosis stages were category according to T2DM and sex.

3.

In Table 2, characteristics of patients was category according to T2DM, obesity, sex, and age <50 years or ≥50 years.

Panel A values is equal to NAFLD-Serum in Table 1, a duplicate content. 

4.

Tables 1 and 2 should be provided the p values.

Why is the age category to <50 years and ≥50 years?

Was the age category provided in Figures?

5.

In Figure 1 (Study design), NAFLD-Serum (n=421,021) is recruited T2D in NAFLD patients (n=254602) before vs.166419 during COVID-19 pandemic

Figure 8. A2M, ApoA1 and haptoglobin levels "before and during" the COVID-19 pandemic (n=254602).

Figure 9. A2M levels" before" the COVID-19 pandemic(n=254602).

Figure 10. A2M levels "during" the COVID-19 pandemic(n=166,419).

It is confused for me.

Similarly, Figures 11, 12, 13, and 14 are the same analysis about before and during the COVID-19 pandemic.

Author Response

Reviewer #1.

  1. Abstract: NAFLD-Serum,n=421.021

Table 1: NAFLD-Serum, n= 427,848

It is a different sample sizes.

Thank you for spotting this typo. The correct value is 427,848. The abstract was modified accordingly in the revised version.

  1. In Table 1,

Characteristics of NAFLD-Biopsy and NAFLD subsets was category according to T2DM and Obese. In figures 2, 5 and 6, characteristics of NAFLD-Biopsy was category according to T2DM and sex. It is a different information (need to same category? ).

Similarly, in Figures 3 and 4, fibrosis stages were category according to T2DM and sex.

It's the same information, but different prism. In the figures, one can easily see an important fact about the gender-effect : most of proteins have a different behavior depending on gender. Also, the main subject of this paper is the effect of T2DM on the proteins. So for the second dimension of the figures it was chosen to stratify according to T2DM status.

The figures stratified by obesity are in the supplementary file : 

  • Supplementary Figure 4: ApoA1 in NAFLD-Biopsied patients with obesity
  • Supplementary Figure 5: A2M in NAFLD-Biopsied patients without obesity
  • Supplementary Figure 6: A2M in NAFLD-Biopsied patients with obesity
  • Supplementary Figure 7: ApoA1 in NAFLD-Biopsied patients without obesity
  • Supplementary Figure 8: Hapto NAFLD-Biopsied patients without obesity
  • Supplementary Figure 9: Hapto in NAFLD-Biopsied patients with obesity
  1. In Table 2,

characteristics of patients was category according to T2DM, obesity, sex, and age <50

years or ≥50 years. Panel A values is equal to NAFLD-Serum in Table 1, a duplicate content.

That is correct, sorry for this duplicate. In the revised version, the Table2 was fixed by removing Panel A. Following panels were renamed accordingly.

  1. Tables 1 and 2 should be provided the p values.

4.1. According to reviewer's comment, the following sentences according to p-values were added with a mark (*) for significant difference vs. control subset in the revised Table 1, and with the following sentences the legend section:

 "In the NAFLD-Biopsy subset,  for subjects with T2DM, with or without obesity, all characteristics were different vs. subjects without T2DM and non-obese (controls), except for ApoA1 level and NASH prevalence. In subject with T2DM and non-obesity ApoA1 was lower. In subjects with no-T2DM and obese the advanced NASH prevalence was higher."

"In the NAFLD-Serum subset, all comparisons showed p-value<0.001."

4.2. The tests used the correction according to the multiple comparisons and were specified in the revised methods' section:

"For qualitative variables, the two-sided Dunnett tests versus a control, of Odds Ratio were used for multiple comparisons. The control subset was the "no-T2DM and non-Obese" subset. For quantitative variables, the Dunnett's two-sided multiple-comparison test with control was used. Difference was significant when P-value <0.05."

4.3. Why is the age category to <50 years and ≥50 years?

Indeed, such definition is always arbitrary.

Here we choose 50 years as one cutoff as it was approximately the median of our large NAFLD cohort. We choose also 25 and 75 years of age according to Ritchie et al works and evidence based studies on protein senescence and degradation. Native A2M normal levels are higher before 25 years of age and the prevalence of degraded proteins increase with age, and we choose 75 years of age for the last cutoff.

This permitted to see for A2M the importance of separating the subjects with high concentration of the native protein before 25 years old and for all the 3 proteins, the importance of degradation after the age of 75 years.

According to the reviewer's selection we added in the revised method section the following sentences:

"The native protein level of A2M is higher before 25 years of age and the prevalence of degraded proteins increase with age, we choose arbitrarily 4 age-based categories with 25 years long: 0-25, 25-50, 50-75 and 75+ years." 

4.4. Was the age category provided in Figures?

In all figures with age-based X axis, 3 vertical black dotted lines display the 4 age categories (0-25, 25-50, 50-75 and 75+) 

According to the reviewer's selection we added in the revised version, for each Figures with the age-based X axis, the following sentences in the corresponding legend:

"Three vertical dotted lines mark the four age groups (before 25, between 25 and 50, between 50 and 75 and above 75 years old)."

  1. In Figures

Thanks to the accurate comments of the reviewer, the errors and the confusing number of subjects, in all figures were checked and fixed in the revised Figure 1 and in the legends of several others.

5.1. Figure 1

(Study design), NAFLD-Serum (n=421,021) is recruited T2D in NAFLD patients

(n=254,602) before vs. 166,419 during COVID-19 pandemic

The correct numbers are 254,602 before COVID-19 (from 2006 to 2019) and 173,246 during the pandemic (Jan 2020-Feb 2022).

According to the reviewer's comments, in the revised Figure 1, the last paragraph third column, fourth raw, has been fixed:

Comparison of proteins levels

according to T2D

in NAFLD patients

254,602 before (2006-2019)

vs.173,246 during

COVID-19 pandemic (2020-Feb 2022)

5.2. Figure 8. A2M, ApoA1 and haptoglobin levels "before and during" the COVID-19 pandemic (n=254602).

Figure 9. A2M levels" before" the COVID-19 pandemic(n=254602).

Figure 10. A2M levels "during" the COVID-19 pandemic(n=166,419).

It is confused for me.

The reviewer comment is right. The number of subjects was wrong for Figure 8 due to copy/paste. The correct value is n=135,911. This is the number of patients included from January 22nd 2020 (start of death count due to Covid-19 in the USA) to February 1st 2022.

The number of subjects in Figure 9 and 10 was correct. 

According to the reviewer's comment, the revised legend of Figure 8, first sentence was modified:

"Figure 8. A2M, ApoA1 and haptoglobin levels before (year 2019) and during the COVID-19 pandemic. Impact of COVID-19 waves on A2M, ApoA1 and haptoglobin level differences with normal values before and during COVID-19 pandemic in obese and non-obese NAFLD-Serum subjects followed for metabolic liver risk (n=135,911)."

5.3. Similarly, Figures 11, 12, 13, and 14 are the same analysis about before and during the COVID-19 pandemic.

We agree that indeed it was confusing.

Figure 8 showing the dynamic of the protein concentration before and after the pandemic, show the data only for the year 2019, before the pandemic, n=135,911. 

The other figures 11-14 show the data using the whole dataset, from 2006 to 2019.

Figures 9, 11 and 13 show the data before the pandemic using data since 2006 (n=254,602).

Figures 10, 12 and 14 show the data during the pandemic (n=173,246).

According to the reviewer's comment, the revised legends, first sentence of Figures 10, 12 and 14, were modified:

"Figure 10. A2M levels during the COVID-19 pandemic. Impact of T2DM on A2M concentration during SARS-CoV-2 pandemic in NAFLD-Serum patients followed for metabolic liver risk (n=173,246)."

"Figure 12. ApoA1 levels during the COVID-19 pandemic. Impact of T2DM on ApoA1 concentration during SARS-CoV-2 pandemic in NAFLD-Serum patients followed for metabolic liver risk (n=173,246)."

Figure 14. Haptoglobin levels during the COVID-19 pandemic. Impact of T2DM on Hapto concentration during SARS-CoV-2 pandemic in NAFLD-Serum patients followed for metabolic liver risk (n=173,246).

Reviewer 2 Report

The paper “Clinical interest of serum alpha-2 macroglobulin, apolipoprotein A1 and haptoglobin, in patients with non-alcoholic fatty liver disease, with and without type-2 diabetes, before or during Covid-19." by Poynard et al. is retrospective non-interventional epidemiological study.

The article is well written, though some sentences should be rephrased. The paper has a good design. The article is logically divided into sections and subsections. The work has a good degree of novelty and of good interest to the readers.

Comments:

  • the introduction is too extensive with 75 references. Is this an article or a review? Moreover, the introduction should briefly introduce to aim and scope of the study which at a certain point comes out of nowhere. Furthermore, in the introduction section it should not be reported the results of the study (line 188-196) as they will be discussed after.
  • In contrast with the extended introduction, discussion section, which is the most important in a study, is poorly discussed. This section MUST be improved. Moreover, line 660-661 “The recent increase in the knowledge of these proteins as well as their easy measurement should improve the confidence of clinicians regarding their performance.” should be further discussed also underlining what clinical implications could represent.
  • Self-citation: I can understand that the authors may have performed other studies or reviews on the discussed topics (one still in press), however, there is a limit to decency. I will report all the self-citation detected for the main authors in order of appearance in the manuscript: Deckmyn 12, Poynard 24, Bedossa 10, Paradis 6, Peta 6, Pais 8, Ratziu 17, Thabut 4, Brzustowski 0, Gautier 0, Cacoub 0, Valla 0.

I was really tempted to reject the article for all the aforementioned reasons, but I do believe that the data presented are original and worth of being published. The article MUST be totally revised.

Author Response

Reviewer #2

The paper “Clinical interest of serum alpha-2 macroglobulin, apolipoprotein A1 and haptoglobin, in patients with non-alcoholic fatty liver disease, with and without type-2 diabetes, before or during Covid-19." by Poynard et al. is retrospective non-interventional epidemiological study.

The article is well written, though some sentences should be rephrased. The paper has a good design. The article is logically divided into sections and subsections. The work has a good degree of novelty and of good interest to the readers.

Comments:

  1. The introduction is too extensive with 75 references. Is this an article or a review?

We agree with the reviewer's comment for a standard original paper in a standard Journal of this level, our introduction is too extensive.

However, in the present context of use, a special issue on Pathological Mechanisms in Diabetes, we try not submit a review, but a very specific original study on the Clinical interest of serum alpha-2 macroglobulin, apolipoprotein A1 and haptoglobin, in patients with non-alcoholic fatty liver disease, with and without type-2 diabetes, before or during Covid-19.

Indeed, we followed the recommendations of the editors of Biomedicines for the introduction:

1.1 Briefly place the study in a broad context and highlight why it is important.

We used 17 lines.

1.2. Purpose of the work and its significance.

We used 10 lines

1.3. The current state of the research field should be carefully reviewed and key publications cited.

Here, we do think that this part of the introduction in a special issue, deserve a complete review of the literature, as the references to the 3 liver proteins are probably unknown by most of the readers. This is also very particular to this topic as the 3 proteins had specific and clinically significant associations with four frequent diseases T2DM, obesity, liver and COVID-19.

Here, we have also the chance not a have a formal restriction in the total number of words.

These points deserve a longer introduction than in a standard article not part of a special issue.

  1. Moreover, the introduction should briefly introduce to aim and scope of the study which at a certain point comes out of nowhere.

As stated in point 2.1, we introduced in 10 lines the purpose of the work and its significance:

" Therefore, these serum proteins, easy to assess, could be used as biomarkers of the risk of SARSCoV-2 infection, particularly in patients with T2D and NAFLD, a large part of the global population, which is an unmet need [9-10].

The epidemiological purpose of the work was to assess the impact of T2D on these proteins according to the main confounders: age, gender, obesity, and the severity of the three liver features, fibrosis, NASH inflammatory activity and steatosis."  

The clinical purpose of the work was to prevent misinterpretation, false positives and negatives, of the serum level of such major serum proteins, observed in patients cumulating metabolic liver disease, T2D and SARS-CoV-2. More and more new proteins will be included in multivariate diagnostic/prognostic tests and should be analyzed similarly.

   In simple terms, a serum level of A2M at 3.8 g/L can be observed in an asymptomatic girl of 18 year of age or in an obese 60-year-old man with cirrhosis."

  1. Furthermore, in the introduction section it should not be reported the results of the study (line 188-196) as they will be discussed after.

Contrarily to the statement of the reviewer, the editors of Biomedicines clearly recommend at the end of the introduction to:

"Finally, briefly mention the main aim of the work and highlight the principal conclusions."

That is why, the following 16 lines were included in the introduction:

"First, the aim was to standardize the three protein values according to age and sex, two major confounding factors, using published normal values from a healthy general population in the USA, called here the "Control-Population" cohort. Second, we assessed the impact of T2DM on these proteins according to the main metabolic liver features of fibrosis and inflammation (NASH) and steatosis without inflammation, stratified by obesity. We used NAFLD patients with liver biopsies who were centralized and analyzed using the validated scoring systems (SAF), called the "NAFLD-Biopsy" cohort. Third, we assessed the impact of T2DM on these proteins in NAFLD-Serum patients followed before and during the COVID-19 pandemic, adjusted according to the confounders of age, sex, and obesity.

We found that in patients at risk of NAFLD, the impact of T2DM on these three proteins should be studied not only after standardization according to age and sex, but also after stratification by obesity. In the two cohorts, the three proteins levels were significantly different than the normal values. In patients at risk of NAFLD without T2DM, A2M was increased, ApoA1 was decreased, and haptoglobin was increased. In patients with both NAFLD and T2DM, these significant differences were magnified. During SARS-CoV-2 infection, this population acquire a third factor of decreasedApoA1 and increased haptoglobin. These results were in line with the independent diagnostic and prognostic values of ApoA1 in COVID-19."

  1. In contrast with the extended introduction, discussion section, which is the most important in a study, is poorly discussed. This section MUST be improved. Moreover, line 660-661 “The recent increase in the knowledge of these proteins as well as their easy measurement should improve the confidence of clinicians regarding their performance.” should be further discussed also underlining what clinical implications could represent.

We agree with the reviewer' s comment that the previous discussion must be improve to be at the level of the introduction.

According to his comments we tried in the revised discussion to improve particularly the clinical interest, diagnostic and prognostic, of these proteins in patients with T2DM, including their limitations.

  1. Self-citation: I can understand that the authors may have performed other studies or reviews on the discussed topics (one still in press), however, there is a limit to decency. I will report all the self-citation detected for the main authors in order of appearance in the manuscript: Deckmyn 12, Poynard 24, Bedossa 10, Paradis 6, Peta 6, Pais 8, Ratziu 17, Thabut 4, Brzustowski 0, Gautier 0, Cacoub 0, Valla 0.

Thank you to the reviewer for this direct count of our self-citations. As an old neo-narcissist,  many friends and dedicated and selfless reviewers like the reviewers #1 and #2, thought that I confuse biography and bibliography.

Here, I understand and respect the comment of the reviewer on the limit of decency.

As a methodologist I appreciate also the statistical obvious correlation between the rank of authorship and the number of self-citation.

 Therefore I will shortly give few arguments for the interest of these citations.

5.1. The in press article (reference #10) is now available: PMID: 35174366 doi: 10.1016/j.gastha.2021.12.009. Online ahead of print.

5.2. Valla is a false negative with 2 citations, indeed cited as senior author in reference #32 and #42.

5.3. Cacoub is also a false negative with 2 citations, indeed cited as senior author in reference #9 and #10.

5.4. Gautier is also a false negative with 1 citation, indeed cited as senior author in reference #32.

5.5. Brzustowski is also a false negative with 1 citation, indeed cited as senior author in reference #32 and #42.

5.6. More seriously, we have not selected the references in order to increase our Hirsh Index, but according to the specificity of the references cited in the state of the art on the key words: A2M-ApoA1-Hapto-T2DM-NAFLD-NASH-Obesity-Fibrosis-Activity-Steatosis-COVID19COVID. Out of 107 references, we cited 24 references of our group, that is 22% of the total. These 24 references represented only 4% than the 600 references of my group published mostly on these proteins. I honestly think that these studies deserve to be cited.

 5.7. According to the reviewer #2 comments we have rewritten the discussion, and added 8 references, without self-citation.

  1. Discussion

Here, the impact of T2DM on the serum levels of three proteins in patients at risk of NAFLD was assessed according to eight major confounding factors: age, sex, obesity, NAFLD histological liver features (fibrosis, NASH steatosis), and SARS-CoV-2 infection. Such a study has never been performed before, despite the rationale and evidence base of such correlations [1-73,92].

Overall, we found in patients at risk of NAFLD without T2DM that the levels of the three proteins were significantly different than normal values, with increased A2M, decreased ApoA1, and increased haptoglobin. In patients with both NAFLD and T2DM, these significant differences were magnified. Furthermore, in cases of SARS-CoV-2 infection, this population had a third factor of decreased ApoA1 and increased haptoglobin. These results are in line with the independent diagnostic and prognostic values of ApoA1 and haptoglobin, combined with A2M in COVID-19 [9-10].

In patients at risk of NAFLD, several multivariate tests used at least one of these three proteins for the non-invasive diagnosis or prognosis of liver features, as follows: A2M (FibroTest/FibroSure, Hepascore, FibroMeter V2G and V3G, and NIS4 [33-45,93-95]), ApoA1 (FibroTest/FibroSure, Chunming score, Shukla score) [31,96,97], and haptoglobin (FibroTest/FibroSure Fuc-Hpt-Mac2bp) [33-45,98]. 

The recent increase in the knowledge of these proteins as well as their inclusion in multivariate biomarkers deserve to discuss their clinical interest and limitations, including their respective risks of false positive and false negatives.

4.1. A2M variability in patients at risk of NAFLD

4.1.1. Strengths

        The variability of A2M according to age in comparison with expected normal values, was retrieved both in NAFLD-Biopsy and NAFLD-Serum cohorts.

    The normal values were highly different according to the four age periods, decreasing rapidly between 5 to 25 years of age, almost stable between 25 to 50, and re-increasing slowly up to more than 75 years of age. These results support both the hypothesis of a preventive role of A2M in SARS-CoV-2 infection in children up to the age of 25 years, and also the aging damage role of A2M after the age of 50 years.

Here, A2M in T2DM was significantly increased at an earlier time point (Figure 2 and Figure 9) than in non-T2DM. This is in line, not only with the role of A2M as a regulator of the extracellular matrix [26-41] and the well-known positive association of A2M in the extracellular matrix of liver fibrosis (Table S1) [19-41], but also with the early damage of proteins associated with the very early presence of T2DM [19,27-29,99]. In men with T2DM of the NAFLD-Serum cohort, A2M increased as soon as the age of 40 years, reaching a mean increase of 0.88 g/L between 50 to 75 years of age (Figure 9).

  4.1.2. Limitations

In NAFLD-Biopsy cohort the sample size was too small to assess the kinetics before the age of 25 years, and therefore to assess the association with liver features. However, thanks to the large sample size of the NAFLD-Serum cohort, the sample size of patients at risk before the age of 25 years was sufficient to compare curves of A2M in patients with T2DM2 vs. normal expected values.

Because of the biopsy, several unexpected results required confirmation and further physio-pathological evidence. A2M levels were decreased in comparison with normal values in patients with significant steatosis S2S3 before the age of 50 years and no T2DM (Figure 4). A specific assessment of “good A2M” (tetramer) versus "bad A2M" (monomer) could be a response.

During SARS-CoV-2 studies in non-human primates, we recently observed an unexpected very early decrease in serum A2M 2 days post-infection at the peak of nasopharyngeal viral loads, with a return to baseline values at the seventh day. Similar kinetics of A2M were observed in hospitalized patients with COVID-19 not requiring intensive care [65]. If confirmed, this kinetic could be associated with the rapid consumption or degradation of native A2M during the peak of the acute phase response. Similar early kinetics of A2M levels were observed in hemodialysis patients, with lower levels during COVID-19 vs. healthy controls, and vs. hemodialysis patients without COVID-19 [11].

  4.1.3. Causal relationships

In healthy subjects, the predominant circulating molecular form of A2M is tetrameric, whereas its dimer was detectable in patients with T2DM and high serum levels of A2M [100]. In nephrotic syndrome, serum A2M levels start to rise when a trace amount of albumin is excreted. Hepatic synthesis of A2M is enhanced significantly to replace lost liver-derived proteins in experimental animals and humans, resulting in a net increase in its serum levels [101-102]. Because of this risk of the confounding factors, we checked the absence of correlations (logistic regression) between the presence of obesity and kidney function assessed by CKD index and A2M levels adjusted by age and sex in the T2DM cohort post-hoc. There was no significant association between A2M and kidney function (Table S2). [103].

4.2. ApoA1

4.2.1. Strengths

Our epidemiological results support the causal relationship between low ApoA1 as an independent risk factor of infection by SARS-CoV-2 in NAFLD patients with additive risks of T2DM and obesity [2-10,13,104].

In NAFLD-Biopsy cohort ApoA1 levels were already significantly decreased (−0.30 g/L) in women with T2DM compared with non-T2DM patients from the age of 30 years (Figure 5), whatever the confounders (Figure S2, Figure S3, Figure S4). This early decrease in patients with biopsy was retrieved in the NAFLD-Serum cohort with a mean decrease of 0.20g/L (Figure 5). In men, the ApoA1 decrease compared with normal values was also significant but much less than in women where the levels increase significantly from birth to 60 years of age.

The association of T2DM with lower ApoA1 (P<0.001) in women with histologically significant NASH, grades A2A3, is original and in line with other more severe liver diseases with high grades of necro-inflammatory inflammation, such as severe alcoholic hepatitis, which was described in previous studies [105-106].

We confirmed here in the NAFLD-Serum cohort the previous findings [9], that before (Figure 11) and during (Figure 12) COVID-19, T2DM was associated with a decrease in ApoA1 compared with non-T2DM patients, in both in men and women. For the first time to our knowledge, we observed that this decrease in ApoA1 was lower in obese vs. non-obese, both before and during Covid-19 (Figure 8).

4.2.2. Limitations

    The present results were limited by the absence of adjustments on confounding factors associated with serum ApoA1 levels, such as dietary folate, physical exercise and vitamin C [107] and alcohol intake both before [108] and during [109] COVID-19. 

4.2.3. Causal relationships with clinical endpoints

    Numerous experimental results explained how ApoA1 may lose its functionality in many inflammatory and pathological conditions, including T2DM, liver diseases, obesity and COVID-19 [49]. ApoA1 is secreted by the liver (about 70%) and the intestine (about 30%). This lipid free ApoA1 interacts with the ATP-binding cassette transporter A1 (ABCA1) on peripheral cells, leading to the transfer of cholesterol and cellular phospholipids from the cell membrane to ApoA1. T2DM, liver diseases, obesity and COVID-19 can impaired the biogenesis of ApoA1 as well as its post translational modifications, including oxidation, carbamylation and glycation.

    Several recent studies separating the roles of HDL main components, such as HDL2, HDL3 and ApoA1 (or according to the HDL size), suggested that the causal relationships between serum levels of ApoA1 with clinical endpoints such as cardio-vascular events [13] or risk of severe COVID-19 [14] or glycosylated hemoglobin [111] were easier to proof, than using the HDL overall levels.     

    Finally a recent work demonstrating a new source of ApoA1 for the liver through the portal vein directly from the intestine opens up many possible causal mechanisms with the risks of T2DM, liver diseases, obesity and COVID-19 [65]. HDL3 produced by the intestine protected the liver from the inflammation and fibrosis observed in a variety of mouse models of liver injury that parallel clinically relevant conditions in humans, including surgical resection of the small bowel, alcohol consumption, or high-fat diets. 

4.3. Haptoglobin

4.3.1. Strengths

The present results underlined the performance of haptoglobin as a sensitive biomarker of inflammation in chronic liver disease. Thus far, the clinical utility of serum haptoglobin was its prognostic value in multivariate blood tests when its levels decreased, which was observed mostly in advanced fibrosis stages [31].

Here in patients at risk of NAFLD, haptoglobin levels were positively associated with male sex, T2DM, and obesity, and surprisingly with the grade of steatosis in non-obese men, in univariate and multivariate analyses. These associations are in line with a chronic inflammation profile in these patients before the onset of histological NASH [66-67,69,72,74].

During SARS-CoV-2 studies in non-human primates, we recently observed that haptoglobin levels, in comparison with CRP, had the same early increase 2 days after infection, but remained more consistently elevated for at least 10 days post-infection [10]. Recent human studies underlined the clinical interest in haptoglobin in SARS-CoV-2 infection [16-18], including one that described the normal distribution of haptoglobin versus the bimodal distribution of CRP [16].

 4.3.2. Limitations

           The present results were limited by the absence of adjustments on confounding factors associated with serum haptoglobin levels, such as haptoglobin polymorphism, inflammatory bowel disease, hemolysis, iron deficiency and exercise.

 4.3.3. Causal relationships with clinical endpoints

           Haptoglobin increase is a validated direct early consequence the acute phase reaction. However, as for ApoA1, serum haptoglobin variability observed in T2DM, obesity, and COVID-19 could be also directly due to the intestine, such as changes in the endothelial permeability. Higher serum level of haptoglobin was observed in obese patients with increased jejunal permeability revealed by lipid challenge and linked to inflammation and T2DM [113]. After an lipid load, haptoglobin was two-fold higher in obese patients compared to non-obese controls and correlated with systemic and intestinal inflammation. Lipid-induced permeability was associated with the presence of T2DM and obesity. In such correlations, the mechanisms explaining the variability of serum haptoglobin could be both genetic defects (such as Gata6 gene and decrease of zonulin the pre-haptoglobin protein), and specific environmental factors (such as high-fat-diet, alcohol, fiber-deprived diet, bacterial or viral infection and medication exposure) known to contribute to break the intestinal barrier balance and promote gut dysbiosis [114].   

4.4. Methodological limitations

The main limitations of the present study are those of the epidemiological study in the non-interventional cohort, which identified significant associations but with multiple tests, many confounding factors, several risks of colinearity, and few direct proofs of causality. 

Furthermore, the impact of T2DM on the three proteins according to SARS-CoV-2 infection was indirectly estimated in a large population at risk of NAFLD, but without direct virological markers. We retrieved the same kinetics of decreased ApoA1 during the successive waves of SARS-CoV-2 infection in comparison with those in 2019, but new confounders could appear such as decreased physical exercise and increased tobacco and/or alcohol consumption [108-109]. However, concerning ApoA1, an increase in alcohol consumption would have been associated with an increase in ApoA1 [55-56,86] in the NAFLD-Serum cohort, which included 78% of subjects without advanced fibrosis [9]. The body weights were also similar (88.5 kg) in 2019 and 2020 [9].

In epidemiological studies of patients with T2DM, several variable and bias factors remained possible candidates, such as the definition of T2DM (clinical definition by diabetologists or fasting glucose) and the number of treatments of protein levels [84]. In the meantime, these variable factors should be discussed in the context of the use of such blood tests components. Here the results of the Control-Population are derived from a relatively homogeneous Caucasian population, and the findings may not be applicable to other ethnic groups. The NAFLD-Biopsy cohort was limited by the selection of patients who accepted a liver biopsy in tertiary centers and their enrollment according to abnormal ALT or the presence of steatosis at ultrasonography. The NAFLD-Serum cohort was limited by the absence of ethnic origin information and the few available clinical characteristics, including age, sex, BMI, ApoA1, A2M, haptoglobin, liver function tests, fasting glucose, total cholesterol, and triglycerides.

Finally, forty years ago, clinicians were not using A2M, ApoA1 was interesting for predicting cardiovascular diseases, and haptoglobin was mostly used for the diagnosis of hemolysis. Nowadays, these proteins are widely prescribed in multivariate noninvasive tests for the diagnosis of liver diseases. However, these three proteins without colinearity between them, but with ubiquitous functions better understood, should permit the construction of better multivariate tests in metabolic diseases cumulating risk of liver diseases and risk of severe infections such as T2DM, NAFLD and Covid-19.

  1. I was really tempted to reject the article for all the aforementioned reasons, but I do believe that the data presented are original and worth of being published. The article MUST be totally revised.

We sincerely thanks the reviewer for his time and clear comments. The revised version is certainly better after his suggestions.

Round 2

Reviewer 1 Report

All the concerns have been answered.